# Simultaneous Statistical Inference for Off-Policy Evaluation in Reinforcement Learning

**Tianpai Luo**[*]  **Xinyuan Fan**[*]  **Weichi Wu**[†]
Department of Statistics and Data Science
Tsinghua University
Beijing 100084, China
{ltp21, fxy22}@mails.tsinghua.edu.cn, wuweichi@tsinghua.edu.cn

## Abstract

This work presents the first theoretically justified simultaneous inference framework for off-policy evaluation (OPE). In contrast to existing methods that focus on point estimates or pointwise confidence intervals (CIs), the new framework quantifies global uncertainty across an infinite or continuous initial state space, offering valid inference over the entire state space. Our method leverages sieve-based Q-function estimation and (high-dimensional) Gaussian approximation techniques over convex regions, which further motivates a new multiplier bootstrap algorithm for constructing asymptotically correct simultaneous confidence regions (SCRs). The widths of the SCRs exceed those of the pointwise CIs by only a logarithmic factor, indicating that our procedure is nearly optimal in terms of efficiency. The effectiveness of the proposed approach is demonstrated through simulations and analysis of the OhioT1DM dataset.

## 1 Introduction

Off-policy evaluation (OPE) is a fundamental topic in reinforcement learning (RL), aiming to assess the performance of a target policy using data collected under a different behavior policy, before adopting the target policy in practice. For this purpose, much effort has been made on the statistical inference of the value of the target policy, including obtaining an accurate estimate and valid confidence intervals for quantifying the uncertainty. See Uehara et al. (2022) for a comprehensive review.

In many real-world applications, such as healthcare (Murphy et al. 2001, Matsouaka et al. 2014, Shi, Lu & Song 2020), ridesharing (Xu et al. (2018)), and autonomous driving (Sallab et al. (2017)), it is often necessary to evaluate a policy across a range of initial states. For instance, in the OhioT1DM dataset (Marling & Bunescu (2020)), each patient begins in a different state of continuous glucose monitoring (CGM) blood glucose levels and self-reported life events. The evaluation of a potentially effective off-policy must be conducted without direct deployment, and it requires quantification of uncertainties across multiple initial states. However, constructing pointwise confidence intervals (CIs) for each state with Bonferroni correction inflates the overall significance level, which is well-known as the multiple testing problem. The inflation becomes especially pronounced when the state space is infinite, for example, $\mathbb{R}$. To address this, we consider the following question:

*Is it possible to simultaneously quantify the uncertainty of off-policy value estimators over the entire state space?*

---

[*]Equal contribution.
[†]Corresponding author.

39th Conference on Neural Information Processing Systems (NeurIPS 2025).

In this paper, we provide an affirmative answer. Specifically, this can be achieved by constructing simultaneous confidence regions (SCRs) that cover the whole value functions at a given significance level.

## 1.1 Related work

Existing methods for statistical inference in reinforcement learning can be categorized into three categories: (i) Direct estimation: This approach constructs CIs by directly learning the system dynamics or Q-function under the target policy. The estimations include kernel-based Q-function methods (Feng et al. (2020)), batch learning (Le et al. (2019)), and sieve estimation methods (Chen (2007), Shi et al. (2021) or equivalently, called linear function approximation Sutton et al. (2008), Lagoudakis (2017)). (ii) Importance sampling: This method re-weights the observed rewards with the density ratio of the target and behavior policies. Bootstrap methods, concentration inequalities, and empirical likelihood-based methods have been applied to construct CIs for importance sampling estimators (Thomas et al. (2015), Hanna et al. (2017), Dai et al. (2020)). (iii) Double reinforcement learning (DRL): This framework combines the first two for more robust and efficient value evaluation (Jiang & Li (2016),Thomas & Brunskill (2016), Jiang & Huang (2020)). For instance, Kallus & Uehara (2022) achieves consistent DRL estimation of the value function and computes a marginalized density ratio to build a CI.

While existing methods largely focus on point estimation or pointwise intervals, approaches tailored for a large number of states remain limited. Works such as Duan et al. (2020) and Shi et al. (2021) advanced this direction by constructing confidence intervals not only pointwise (for the value at a given state) but also for integrated value functions under a known reference distribution of initial states. However, both leave important gaps. The asymptotic theory in Shi et al. (2021) establishes validity in large samples but does not provide non-asymptotic error control. In contrast, Duan et al. (2020) supports finite-sample inference, but its confidence bounds are conservative. Neither framework provides the simultaneous inference that is uniformly valid across all states. Our work addresses these gaps by developing a framework that enables distribution-free, asymptotically correct inference for the value function at any state simultaneously, while also delivering finite-sample guarantees through the non-asymptotic bound obtained from the Gaussian approximation.

## 1.2 Contributions

In this paper, we propose a novel framework for constructing asymptotically correct SCRs for the OPE. To the best of our knowledge, this is the first work to introduce a simultaneous statistical inference framework in policy evaluation of RL. Our method shares a similar spirit to Q-learning, which estimates the state-action value function (Q-function) under the target policy. The estimation of Q-function is achieved by the linear function approximation (i.e., sieve method). Our key contributions are as follows:

1. We establish a convex Gaussian approximation result for the sieve estimation of the Q-function. This approximation enables us to characterize the distribution of the sieve estimator over arbitrary convex sets, thereby facilitating simultaneous inference when the initial state is not fixed. Moreover, the convex Gaussian approximation theory only requires the number of trajectories or decision points to diverge, which naturally allows the infinite-horizon setting. Our theoretical results are built upon non-asymptotic results, which do not involve any convergence from extreme value theory in statistics.

2. Based on the convex Gaussian approximation, we construct an asymptotically correct SCR whose stochastic behavior is depicted by the maxima of a Gaussian random field. The width of the SCR exceeds that of the pointwise confidence intervals only by a logarithmic factor.

3. To implement our methodology, we develop a multiplier bootstrap algorithm for constructing SCRs, which avoids the need to estimate the limiting joint distribution of policy value estimators across different initial states. We further assess the performance of the proposed simultaneous inference framework through both numerical simulations and real data analysis.

The rest of the article is organized as follows. We introduce the model setup in Section 2. In Section 3, we present the construction of SCR based on sieve estimation, convex Gaussian approximation, and the bootstrap algorithm. Simulation studies and real data analysis on the OhioT1DM dataset are

conducted in Section 4. Finally, we conclude our paper in Section 5. All proofs, along with additional simulation results, are given in the supplementary material.

## 2 Preliminaries

Consider a Markov Decision Process (MDP) represented by the tuple $\mathcal{M} = \langle \mathcal{S}, \mathcal{A}, R \rangle$, where $\mathcal{S}$ denotes the state space, $\mathcal{A}$ the action space, and $R : \mathcal{S} \times \mathcal{A} \to \mathbb{R}$ the reward function. In this paper, we assume that $\mathcal{S}$ is a subspace of $\mathbb{R}^d$ with a fixed dimension $d$, and $\mathcal{A}$ is the discrete set $\{0, 1, \ldots, m-1\}$ with a fixed cardinality $m$. Let $(S_{0,t}, A_{0,t}, R_{0,t})$ denote the state-action-reward triplet collected at time $t$. In the MDP framework, the following Markov assumption is imposed:

$$\mathrm{P}(S_{0,t+1} \in \mathcal{B}|S_{0,t} = s, A_{0,t} = a, \{S_{0,k}\}_{k<t}, \{A_{0,k}\}_{k<t}, \{R_{0,k}\}_{k<t}) = \mathcal{P}(\mathcal{B}|s,a), \qquad (2.1)$$

where $\mathcal{P}$ denotes the transition probability kernel, which is time-homogeneous. Additionally, we assume that the conditional mean of the reward $R_{0,t}$ depends only on the current state and action, i.e.,

$$\mathbb{E}(R_{0,t}|S_{0,t} = s, A_{0,t} = a, (S_{0,k}, A_{0,k}R_{0,k})_{k<t}) = \mathbb{E}(R_{0,t}|S_{0,t} = s, A_{0,t} = a) = r(s,a), \quad (2.2)$$

where $r(\cdot)$ is a reward function $r : \mathcal{S} \times \mathcal{A} \to \mathbb{R}$. We note that if the reward $R_{0,t}$ is a deterministic function of $S_{0,t}, A_{0,t}, S_{0,t+1}$, condition (2.2) follows directly from (2.1). Both (2.1) and (2.2) are standard assumptions in the reinforcement learning literature.

Let $\pi(\cdot|\cdot)$ denote a policy which satisfies $\pi(a|s) \geq 0$ for all $s \in \mathcal{S}, a \in \mathcal{A}$, and $\sum_{a \in \mathcal{A}} \pi(a|s) = 1$ for any $s \in \mathcal{S}$. The objective of RL can then be expressed through the following value function:

$$V(\pi; s) = \sum_{t \geq 0} \gamma^t \mathbb{E}^\pi(R_{0,t}|S_{0,0} = s), \qquad (2.3)$$

where the expectation $\mathbb{E}^\pi$ is taken under the rule that actions are selected according to the policy $\pi$, and $\gamma$ refers to a given discount factor, $0 \leq \gamma < 1$.

In this paper, we consider an offline setting where data is pre-collected and can be written as

$$(R_{i,t}, A_{i,t}, S_{i,t}, S_{i,t+1}), \ 0 \leq t \leq T_i, \ 1 \leq i \leq n,$$

where $n$ denotes the number of trajectories, and $T_i$ is the termination time of the $i$-th trajectory. For the sake of brevity, we assume $T_i = T, i = 1, \ldots, n$, and the sample size is denoted as $N = nT$. Our framework only requires that either $T$ or $n$ diverges (namely, $N \to \infty$).

## 3 Simultaneous inference for OPE

In this paper, we shall construct the asymptotically correct SCR for the OPE at significance level $1 - \alpha, \alpha \in (0, 1)$ via finding $C_\alpha$ (which might depend on $N$) such that

$$\lim_{N \to \infty} \mathrm{P}\left(\hat{V}(\pi; s) - C_\alpha L(s) \leq V(\pi; s) \leq \hat{V}(\pi; s) + C_\alpha L(s), \forall s \in \mathcal{S}\right) = 1 - \alpha, \qquad (3.1)$$

where $\hat{V}(\pi; s)$ is the estimated policy values and $L(s)$ is a scaling factor related to the covariance. When only a fixed $s_0 \in \mathcal{S}$ is considered (instead of $\forall s \in \mathcal{S}$), (3.1) reduce to the pointwise confidence interval. Since the state space $\mathcal{S}$ can be continuous and infinite, to achieve asymptotic correct simultaneous coverage, we need to well control the family-wise error rate in contrast with previous pointwise CIs in RL (e.g., Luckett et al. (2020), Shi et al. (2021), Shi et al. (2024)).

Without loss of generality, we focus on *stationary policies* $\pi(\cdot \mid \cdot)$ that do not vary with time $t$. For the justification, we refer to Lemma 1 of Shi, Wan, Song, Lu & Leng (2020) and proof of Theorem 6.2.10 in Puterman (1994). To enable simultaneous confidence inference, we impose three main assumptions, which are adopted from the literature on pointwise inference (e.g., Shi et al. (2021)). The detailed assumptions and illustrations are listed as (A1)–(A3) in Section 3.2.

### 3.1 Q-learning with linear function approximation

We adopt a Q-learning approach to develop valid inference procedures for both deterministic and random policies. The Q-function under a policy $\pi$ is defined as

$$Q(\pi; s, a) = \sum_{t \geq 0} \gamma^t \mathbb{E}^\pi(R_{0,t}|S_{0,0} = s, A_{0,0} = a). \qquad (3.2)$$

Under conditions (2.1) and (2.2), the Q-function satisfies the Bellman equation:

$$Q(\pi; s, a) = r(s, a) + \gamma \mathbb{E}^{\pi} \left[ Q\left(\pi; S_{0,1}, A_{0,1}\right) \mid S_{0,0} = s, A_{0,0} = a\right]. \quad (3.3)$$

We consider the linear function approximation for learning the Q-function. Let $\Phi_1(s), \Phi_2(s), \ldots, \Phi_K(s)$ be a collection of $K$ basis functions and $\boldsymbol{\Phi}(s) = (\Phi_1(s), \ldots, \Phi_K(s))^{\top}$. We approximate $Q(\pi; s, a)$ based on a linear combination of the basis functions, i.e.,

$$Q(\pi; s, a) \approx \boldsymbol{\Phi}(s)^{\top} \beta_{\pi, a}^*, \forall s \in \mathcal{S}, a \in \mathcal{A}. \quad (3.4)$$

Related approximation results are presented in Section F.1 of the supplementary materials, assuming that $Q(\pi; \cdot, a)$ belongs to a Hölder space of smoothness $p$ for any policy $\pi$ and action $a \in \mathcal{A}$. This condition holds under standard assumptions on the transition probability $\mathcal{P}$ and a smooth reward function $r(s, a)$ (see Section F.1 for details). The basis functions can be chosen from orthogonal splines, Legendre polynomials, or wavelets, forming a sieve basis commonly used in sieve estimation (Chen 2007, Huang 1998, Cohen et al. 1993, Timan 2014).

By (3.3) and (3.4), the $mK$-dimensional vector $\beta_{\pi}^* = (\beta_{\pi,1}^{*\top}, \ldots, \beta_{\pi,m-1}^{*\top})^{\top}$ satisifies

$$\mathbb{E} \left\{ R_{i,t} + \gamma \sum_{a \in \mathcal{A}} \Phi(S_{i,t+1})^{\top} \beta_{\pi,a}^* \pi(a|S_{i,t+1}) - \Phi(S_{i,t})^{\top} \beta_{\pi,a'}^* \right\} \Phi(S_{i,t}) \mathbb{I}(A_{i,t} = a') = 0, \quad (3.5)$$

for all $a' \in \mathcal{A}$. Denote $\xi_{i,t} = \xi(S_{i,t}, A_{i,t}), U_{\pi,i,t} = U_\pi(S_{i,t})$ where

$$\xi(s, a) = \left\{ \Phi(s)^{\top} \mathbb{I}(a = 0), \Phi(s)^{\top} \mathbb{I}(a = 1), \ldots, \Phi(s)^{\top} \mathbb{I}(a = m - 1) \right\}^{\top},$$

$$U_\pi(s) = \left\{ \Phi(s)^{\top} \pi(0|s), \Phi(s)^{\top} \pi(1|s), \ldots, \Phi(s)^{\top} \pi(m-1|s) \right\}^{\top}.$$

Then (3.5) reduces to $\mathbb{E}\xi_{i,t}(R_{i,t} + \gamma U_{\pi,i,t+1}^{\top} \beta_{\pi}^* - \xi_{i,t}^{\top} \beta_{\pi}^*) = 0$, and $\beta_{\pi}^*$ can be estimated by

$$\hat{\beta}_\pi = \hat{\Sigma}_\pi^{-1} \left( \frac{1}{\sum_i T_i} \sum_{i=1}^{n} \sum_{t=0}^{T_i-1} \xi_{i,t} R_{i,t} \right), \quad (3.6)$$

where $\hat{\beta}_\pi = (\hat{\beta}_{\pi,1}^{\top}, \ldots, \hat{\beta}_{\pi,m-1}^{\top})^{\top}$ and

$$\hat{\Sigma}_\pi = \frac{1}{\sum_i T_i} \sum_{i=1}^{n} \sum_{t=0}^{T_i-1} \xi_{i,t} \left( \xi_{i,t} - \gamma U_{\pi,i,t+1} \right)^{\top}. \quad (3.7)$$

Consequently, the value for policy $\pi$ can be estimated by

$$\hat{V}(\pi; s) = \sum_{a \in \mathcal{A}} \pi(a|s) \hat{Q}(\pi; s, a) = \sum_{a \in \mathcal{A}} \pi(a|s) \Phi(s)^{\top} \hat{\beta}_\pi. \quad (3.8)$$

By (3.4), we have $\hat{V}(\pi; s) - V(\pi; s) - \sum_{a \in \mathcal{A}} \pi(a|s) \Phi(s)^{\top} \hat{\theta}_\pi = O(\epsilon_K)$ where $\hat{\theta}_\pi = \hat{\beta}_\pi - \beta_{\pi}^*$ and $\epsilon_K = \max_{a \in \mathcal{A}} \sup_{s \in \mathcal{S}} |Q(\pi; s, a) - \Phi(s)^{\top} \beta_{\pi,a}^*|$. By Chen (2007), there exists $\beta_{\pi}^*$ such that $\epsilon_K = O(K^{-p/d})$ when the Q-function lies in a $d$-dimensional space with Hölder smoothness $p$.

### 3.2 Convex Gaussian approximation

In this section, we establish a general convex Gaussian approximation theory for learning the distribution behavior of $\hat{\theta}_\pi = \hat{\beta}_\pi - \beta_{\pi}^*$ for all Euclidean convex sets in $\mathbb{R}^{mK}$. To allow $K$ to diverge, we apply convex Gaussian approximation theorem (Fang (2016), Fang & Koike (2024)), which supports moderately high-dimensional scenarios. We consider the state $s$ within a compact region $\mathcal{S} \subset \mathbb{R}^d$. For unbounded $\mathcal{S}$, modifications such as introducing a weighting or mapping function are discussed in, e.g., Tjøstheim & Auestad (1994), Huang & Shen (2004), Chen & Christensen (2015). We impose the following assumptions.

(A1) The Markov chain $\{S_{0,t}\}_{t \geq 0}$ has an unique invariant distribution with some density function $\mu(s)$. Denote $\nu_0(s)$ as the probability density function of $S_{0,0}$. The density functions $\mu(s)$ and $v_0(s)$ are uniformly bounded away from 0 and $\infty$.

(A2) Suppose the following (i) and (ii) hold when $T \to \infty$ and (i) holds when $T$ is bounded. (i) $\lambda_{\min}\left[\sum_{t=0}^{T-1} \mathbb{E}\left\{\xi_{0,t}\xi_{0,t}^{\top} - \gamma^2 \boldsymbol{u}_\pi\left(S_{0,t}, A_{0,t}\right) \boldsymbol{u}_\pi^{\top}\left(S_{0,t}, A_{0,t}\right)\right\}\right] \geq T\bar{c}$ for some constant $\bar{c} > 0$, where $\boldsymbol{u}_\pi(x,a) = \mathbb{E}\left\{\boldsymbol{U}_\pi\left(S_{0,1}\right) \mid S_{0,0} = x, A_{0,0} = a\right\}$ and $\lambda_{\min}(\mathbf{M})$ denotes the minimum eigenvalue of a matrix $\mathbf{M}$. (ii) $\{S_{0,t}\}_{t\geq 0}$ is geometric ergodicity in dependence measure.

(A3) Define $\omega_\pi(s,a) = \mathbb{E}\left[\left\{R_{0,0} + \gamma \sum_{a\in\mathcal{A}} \pi(a|S_{0,1})Q(\pi; S_{0,1}, a) - Q(\pi; S_{0,0}, A_{0,0})\right\}^2\right]$. Assume $\omega_\pi(s,a) \geq c_0^{-1}$ and $\mathrm{P}(\max_{0\leq t\leq T-1} |R_{0,t}| \leq c_0) = 1$ for some constant $c_0 \geq 1$.

**Remark 3.1.** *Assumptions (A1)–(A3) are mild assumptions and serve as the minimal requirement for the goodness of the offline dataset to support feasible evaluation. The first condition in Assumption (A1) ensures that the Markov chain would not be trapped in a small subset of the entire space. Moreover, the second condition ensures that every state is possible to be the initial state. Assumption (A2) relaxes the condition on sample size. Previous work (Jiang & Li 2016) requires the number of trajectories $n \to \infty$. (A2) additionally allows fixed $n$, but length $T \to \infty$ when the action variety is sufficiently large on each chain. The geometrical decay is similar with the geometrical ergodic for the Markov chain, which is a technical assumption in theoretical deduction, and is commonly assumed as a weaker requirement of i.i.d. in deriving limit theory. Assumption (A3) requires the reward signal diversity. $\omega_\pi(s,a) \geq c_0^{-1}$ requires that the reward random variable is nondegenerate (not always the same). $\mathrm{P}(\max_{0\leq t\leq T-1} |R_{0,t}| \leq c_0) = 1$ means the rewards are bounded. The detailed definitions of the geometrical ergodicity and dependence measure are presented in Section G of the supplementary materials to save space.*

The following Theorem 3.1 shows that there exists $mK$-dimensional Gaussian random vector $\mathbf{Z}_\pi$ such that probability of $\hat{\theta}_\pi = \hat{\beta}_\pi - \beta_\pi^*$ can be approximated by $\mathbf{Z}_\pi$ over any convex sets.

**Theorem 3.1.** *Denote $\mathbf{Z}_\pi$ as the $mK$-dimensional Gaussian random vector possesses the same covariance structure of $\sqrt{N}\hat{\theta}_\pi$, i.e.,*

$$\mathbf{Z}_\pi \sim \mathcal{N}_{mK}\left(\mathbf{0}, \Lambda_\pi\right), \quad \Lambda_\pi = \mathbb{E}\left\{\hat{\Sigma}_\pi^{-1}\hat{\Omega}_\pi(\hat{\Sigma}_\pi^{\top})^{-1}\right\}, \tag{3.9}$$

*where*

$$\hat{\Omega}_\pi = \frac{1}{N}\sum_{i=1}^{n}\sum_{t=0}^{T_i-1} \xi_{i,t}\xi_{i,t}^{\top}\left\{R_{i,t} + \gamma U_{\pi,i,t+1}^{\top}\hat{\beta}_\pi - \xi_{i,t}^{\top}\hat{\beta}_\pi\right\}^2. \tag{3.10}$$

*Under Assumptions (A1), (A2), and (A3), suppose that $K = o\left(N^{2/7}(\log N)^{-1}\right)$, then we have*

$$\sup_{\mathbb{O}\in\mathfrak{O}} |\mathrm{P}(\sqrt{N}\hat{\theta}_\pi \in \mathbb{O}) - \mathrm{P}(\mathbf{Z}_\pi \in \mathbb{O})| \to 0, \tag{3.11}$$

*where $\mathfrak{O}$ is the collection of all the convex sets in $\mathbb{R}^{mK}$.*

**Remark 3.2.** *Note that the SCR based on estimation $\hat{V}(\pi;s) = \Phi(s)^{\top}\sum_{a\in\mathcal{A}} \pi(a|s)\hat{\beta}_\pi$ can be written as $\cap_{s\in\mathcal{S}}\{\sqrt{N}\hat{\theta} \in \mathbb{O}_{\pi,s}\}$ where*

$$\mathbb{O}_{\pi,s} = \left\{\theta \in \mathbb{R}^{mK} : \left|\Phi(s)^{\top}\sum_{a\in\mathcal{A}} \pi(a|s)\theta\right| \leq L(s)\right\}. \tag{3.12}$$

*$\mathbb{O}_{\pi,s}$ is a convex set since for any $\theta, \theta' \in \mathbb{O}_{\pi,s}$, $\lambda\theta + (1-\lambda)\theta' \in \mathbb{O}_{\pi,s}$ for any $\lambda \in [0,1]$. Therefore, $\cap_{s\in\mathcal{S}}\mathbb{O}_{\pi,s}$ is a convex set and the probability $\mathrm{P}(\cap_{s\in\mathcal{S}}\{\sqrt{N}\hat{\theta} \in \mathbb{O}_{\pi,s}\})$ can be learned by $\mathrm{P}(\cap_{s\in\mathcal{S}}\{\mathbf{Z}_\pi \in \mathbb{O}_{\pi,s}\})$.*

**Remark 3.3.** *Theorem 3.1 provides a higher-order convex Gaussian approximation for $\hat{\theta}_\pi = \hat{\beta}_\pi - \hat{\beta}_\pi^*$ in the OPE estimation error $\hat{V}(\pi;s) - V(\pi;s) = \Phi(s)^{\top}\sum_{a\in\mathcal{A}} \pi(a|s)\hat{\theta}_\pi$. Existing approaches for constructing pointwise confidence intervals typically rely on the central limit theorem, deriving the limiting distribution of the inner product $\Phi(s)^{\top}\sum_{a\in\mathcal{A}} \pi(a|s)\hat{\theta}_\pi$ for each fixed $s \in \mathcal{S}$. However, extending these results from a fixed $s$ to arbitrary $s \in \mathcal{S}$ is nontrivial, as it requires controlling $\Delta_{\mathbb{O}} = \left|\mathrm{P}\left(\sqrt{N}\hat{\theta}_\pi \in \mathbb{O}\right) - \mathrm{P}\left(\mathbf{Z}_\pi \in \mathbb{O}\right)\right|$ for some convex set $\mathbb{O}$ (see Remark 3.2 for details).*

*Regarding the finite-sample properties, we provide the following bound on $\Delta_{\mathbb{O}}$ with respect to the sample size $N$ and the number of basis functions $K$, derived from the proof of Theorem 3.1:*

$$\sup_{\mathbb{O}\in\mathfrak{O}} \Delta_{\mathbb{O}} \leq C \left( \sqrt{K^{\frac{1}{4}} N^{\frac{1}{2}} \pi_N^{1-q} \xi_{K,N}^q} + K^{\frac{1}{8}} N^{\frac{1}{2}} \pi_N^{1-q} \xi_{K,N}^q + K^{\frac{1}{4}} N^{-\frac{1}{2}} \pi_N^3 \log^2 N \right).$$

*From the proof of Theorem 3.1, it follows that the above bound converges to 0 as $N \to \infty$ when $K = o\left(N^{2/7-c}\right)$ for any given $c > 0$.*

By Theorem 3.1, the SCR in (3.1) can be achieved by finding appropriate critical value $C_{\alpha,N} > 0$ (which may depend on $N$) such that

$$1 - \alpha = \mathrm{P}\left\{ \left| \Phi(s)^\top \sum_{a\in\mathcal{A}} \pi(a|s)\mathbf{Z}_\pi \right| \leq C_{\alpha,N} L(s), \forall s \in \mathcal{S} \right\},$$

$$= \mathrm{P}\left\{ \sup_{s\in\mathcal{S}} \left| \frac{\Phi(s)^\top \sum_{a\in\mathcal{A}} \pi(a|s)\mathbf{Z}_\pi}{L(s)} \right| \leq C_{\alpha,N} \right\}. \tag{3.13}$$

The probability in (3.13) involves the supremum of functional linear combinations of the high-dimensional Gaussian vector $\mathbf{Z}_\pi$. In practice, we can approximate $C_{\alpha,N}$ in (3.13) by generating simulations of the Gaussian random vector $\mathbf{Z}_\pi$ and computing the empirical quantile of the supremum. This approach, known as the Gaussian multiplier bootstrap, is detailed in Section 3.3. In theory, we leverage properties of Gaussian processes along with approximation techniques from Sun & Loader (1994) to analyze the desired $C_{\alpha,N}$.

**Proposition 3.2.** *For any two positive real sequences $a_n$ and $b_n$, we write $a_n \asymp b_n$ if there exists constants $0 < c < C < \infty$ such that $c \leq \liminf_{n\to\infty} a_n/b_n \leq \limsup_{n\to\infty} a_n/b_n \leq C$. We write $a_n \lesssim b_n$ ($a_n \gtrsim b_n$) if there exists constant $C > 0$ such that $a_n \leq Cb_n$ ($Ca_n \geq b_n$) for all $n$. Denote matrix*

$$\mathbf{M} =: (\mathbf{M}_1(s),\ldots,\mathbf{M}_d(s)), \quad \mathbf{M}_j(s) =: \frac{\partial}{\partial s_j}\left( \frac{\Phi(s)}{|\Phi(s)|} \right). \tag{3.14}$$

*Under same conditions in Theorem 3.1, if there exists constant $c_0, c_1, c_2, \underline{c} \geq 0$ such that*

$$\sup_{s\in\mathcal{S}} |\nabla\Phi(s)| \lesssim N^{c_1}, \sup_{s\in\mathcal{S}} |\nabla^2\Phi(s)| \lesssim N^{c_2}, \inf_{s\in\mathcal{S}} |\Phi(s)| \gtrsim N^{c_0}, \int_{\mathcal{S}} \lambda_{min}(\mathbf{M}^\top\mathbf{M})\mathrm{d}s \gtrsim N^{\underline{c}}, \tag{3.15}$$

*then we have appropriate $C_{\alpha,N} \asymp \log^{1/2} N$ such that*

$$\lim_{N\to\infty} \mathrm{P}\left\{ \left| \frac{\hat{V}(\pi;s) - V(\pi;s)}{\sqrt{U_\pi(s)^\top \Lambda_\pi U_\pi(s)}} \right| \leq \frac{C_{\alpha,N}}{\sqrt{N}}, \forall s \in \mathcal{S} \right\} = 1 - \alpha. \tag{3.16}$$

*where $\alpha$ is the given significance level and $\alpha \in (0,1)$.*

**Remark 3.4.** *The scaling factor $L(s) = \sqrt{U_\pi(s)^\top \Lambda_\pi U_\pi(s)}$ aligns with the pointwise CIs in Shi et al. (2021) so that we only need to compare the critical value $C_{\alpha,N}$ with that in pointwise CIs. The rates $N^{c_1}, N^{c_2}, N^{c_0}$ in condition (3.15) are mild assumptions which have been frequently used in the literature of sieve nonparametric estimation and inference; see Assumption 4 of Chen & Christensen (2015) and Example 1-2 in Quan & Lin (2024) for more details. The rate $N^{\underline{c}}$ for $\int_{\mathcal{S}} \lambda_{min}(\mathbf{M}^\top\mathbf{M})\mathrm{d}s$ can be derived in practice given basis $\Phi(s)$, which would be verified in Section F.2 of the supplementary materials.*

Proposition 3.2 specifies the essential scale of the width of SCR. In contrast with previous pointwise confidence intervals proposed in Shi et al. (2021), $C_{\alpha,N} \asymp \sqrt{\log N}$ shows that only an additional logarithmic rate $\sqrt{\log N}$ is introduced to extend the pointwise confidence in Shi et al. (2021) to the global region $\mathcal{S}$.

### 3.3 Bootstrap implementation

In the asymptotically correct SCR provided by (3.16), calculating an approximation of $C_{\alpha,N}$ is rather complicated, and the convergence would be slow. We propose the Gaussian multiplier bootstrap algorithm to circumvent these problems and derive a feasible SCR in Algorithm 1.

---

**Algorithm 1** Gaussian multiplier bootstrap for SCR

---

**Input:** Observed data $\{(R_{i,t}, A_{i,t}, S_{i,t}, S_{i,t+1})\}_{0 \leq t \leq T_i, 1 \leq i \leq n}$.

**Step 1:** Calculate $\hat{\beta}_\pi$, $\hat{\Sigma}_\pi$, $\hat{\Omega}_\pi$ according to (3.6), (3.7), and (3.10). Obtain the estimator of the value function

$$\hat{V}(\pi; s) = \Phi(s)^\top \sum_{a \in \mathcal{A}} \pi(a|s)\hat{\beta}_\pi. \tag{3.17}$$

**Step 2:** Generate $mK$-dimensional Gaussian random vector $\mathbf{Z}_\pi^{(b)} \sim \mathcal{N}_{mK}\left(\mathbf{0}, \hat{\Sigma}_\pi^{-1}\hat{\Omega}_\pi(\hat{\Sigma}_\pi^\top)^{-1}\right)$.

**Step 3:** Repeat Step 2 for $B$ times and document the outcomes $\mathbf{Z}_\pi^{(b)}$, $b = 1, \ldots, B$.

**Step 4:** For a given level $\alpha \in (0, 1)$, denote $\hat{q}_{1-\alpha}$ as the $(1-\alpha)$-th sample quantile of

$$\left\{ \sup_{s \in \mathcal{S}} \left| L(s)^{-1} \Phi(s)^\top \sum_{a \in \mathcal{A}} \pi(a|s)\mathbf{Z}_\pi^{(b)} \right| \right\}_{b=1}^{B}.$$

**Output:** $(1 - \alpha)$-th SCR $\hat{V}(\pi; s) \pm \hat{q}_{1-\alpha}L(s)/\sqrt{N}$.

---

It is worth noting that by the convex Gaussian approximation results, Algorithm 1 can yield different asymptotically correct SCRs by modifying the scaling factor $L(s)$. The function $L(s)$ provides flexibility to adjust the relative weighting of states $s \in \mathcal{S}$, where larger $L(s)$ prioritizes tighter confidence bounds for state $s$. For instance, if one is interested in the uncertainty of maximal deviation $\sup_{s \in \mathcal{S}} |\hat{V}(\pi; s) - V(\pi; s)|$, then $L(s) = 1$ can be a convenient choice.

**Remark 3.5** (Computational remarks). *Our procedure is computationally efficient and can, for instance, be executed on a personal laptop. The term $\hat{\beta}_\pi$ in (3.6) is analogous to a least squares estimate and can be computed efficiently. Moreover, Steps 1 and 4 of the bootstrap procedure are linear due to the use of a linear approximation. Overall, the time complexity of our method is $O(NK^2 + K^3 + BK)$ and the space complexity is $O(N + BK)$.*

# 4 Experiments

## 4.1 Simulation studies

In this section, we conduct numerical studies to evaluate the performance of the proposed SCR. Both univariate $(d = 1)$ and multivariate $(d > 1)$ scenarios are considered. The code is available at https://github.com/xinyuanfan01/Simultaneous-Statistical-Inference-for-Off-Policy-Evaluation-in-Reinforcement-Learning.

In our settings, the state vector $S_{0,t}$ may not have bounded support. To address this, we apply a sigmoid transformation, defined as $\text{sigmoid}(S_{0,t}^{(j)}) = \frac{1}{1+\exp(-S_{0,t}^{(j)})}$ for $1 \leq j \leq d$, to obtain features with bounded support. The basis functions are constructed using the tensor product of $K$ Legendre or spline functions. The number of basis functions is determined through cross-validation (Qiu et al. 2021). We put the detailed cross-validation procedure in Section D of the supplement. Moreover, we performed sensitivity analyses and found that both the empirical coverage and the average length of the SCRs are robust to the choice of $K$.

We evaluate the SCRs using two metrics, each computed across 500 independent replications: (i) Empirical Coverage Probability (ECP): The proportion of times the true value function lies within the SCR across multiple simulations. (ii) Average Length (AL): The average width of the SCRs, approximated by averaging the widths at equally spaced grid points. The experiments can be readily conducted on a standard workstation, for example, an Apple M1 machine with 16 GB of RAM running macOS Sonoma.

For the method in Shi et al. (2021) (referred to as SAVE), we apply the Bonferroni correction to adjust the pointwise confidence intervals. For each setting, we compute the SCRs over equally spaced grid points. We emphasize that, in principle, pointwise confidence intervals cannot be naturally extended to SCRs, due to the continuous nature of our state space. Overall, the proposed SCR

achieves coverage close to the nominal level (we set $\alpha = 0.05$), while the Bonferroni-adjusted SAVE results in a coverage rate well above $0.95$.

**(Scenario 1 (univariate).)** Let $\gamma = 0.5, n = 25, 50, 75, T = 30, 50, 70$, and

$$S_{0,t+1} = S_{0,t} + (2A_{0,t} - 1)U_{0,t}, \ R_{0,t} = -\frac{1}{2}\frac{e^{S_{0,t+1}^2} - e^{-S_{0,t+1}^2}}{e^{S_{0,t+1}^2} + e^{-S_{0,t+1}^2}},$$

for $t \geq 0$, where $U_{0,t} \overset{i.i.d.}{\sim} U(0,1)$ and $S_{0,0} \sim U(-2,2)$. We consider a completely randomized behavior policy, i.e., $A_{0,t} \overset{i.i.d.}{\sim}$ Bernoulli$(0.5)$ for $t \geq 0$. The target policy is designed as $\pi(1|s) = 1 - I(s > 0)$. We construct SCRs for $V(\pi, s)$ over the domain $s \in [-2, 2]$. The true value function is approximated from Monte Carlo simulation. We generate 10000 of independent trajectories with initial state being $s_0^i = -2 + 4i/999$ for each $i = 0, \ldots, 999$. Actions are selected according to the target policy. We approximate $V(\pi, s_0^i)$ by taking the average over the 10000 trajectories, and use linear interpolation to approximate $V(\pi, s)$ for $s \notin \{s_0^i, \ i = 0, \ldots, 999\}$.

For Scenario 1, we employ the Legendre basis, and the results for ECPs and ALs are presented in Figure 1. Moreover, we perform the sensitivity analysis by taking $(n, T) = (25, 50), (50, 50)$ as two illustrative examples and examining the results by varying the specification of $K$ over a relatively wide range. The corresponding results are presented in Table S.1 in the supplement. Figure 1 shows

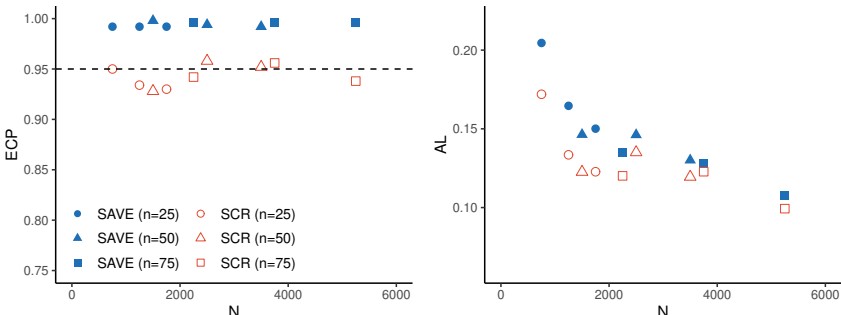

Figure 1: Comparison of the methods based on empirical coverage probability (ECP, left) and average length (AL, right) for Scenario 1.

that SCR consistently achieve the nominal coverage level in various choices of $(n, T)$. In contrast, the Bonferroni-adjusted SAVE method exhibits substantial over-coverage. As $N = nT$ increases, the empirical performance of our method converges more closely to the nominal target. Additionally, the average width of the SCRs produced by SAVE is approximately 20% greater than that of ours, which highlights the improved efficiency of our approach. Table S.1 further shows that our method is robust to the choice of the number of basis functions, enhancing its practical applicability.

**(Scenario 2 (multivariate).)** Let $\gamma = 0.5$,

$$S_{0,t+1} = \frac{3}{4}\begin{pmatrix} 2A_{0,t} - 1 & 0 \\ 0 & 1 - 2A_{0,t} \end{pmatrix} S_{0,t} + z_{0,t}, \ R_{0,t} = S_{0,t+1}^\top \begin{pmatrix} 2 \\ 1 \end{pmatrix} - \frac{1}{4}(2A_{0,t} - 1),$$

for $t \geq 0$, where $z_{0,t} \overset{i.i.d.}{\sim} N(\mathbf{0}, 4\mathbf{I}_2)$ and $S_{0,0} \sim U([-2, 2]^2)$, where the two components are independent. For behavior policy, we consider $A_{0,t} \sim$ Bernoulli$(p_{0,t})$ independently, where $p_{0,t} = 0.5\left(\text{Sigmoid}(S_{0,t}^{(1)}) + \text{Sigmoid}(S_{0,t}^{(2)})\right)$. The target policy is designed as $\pi(1|s) = I(s^{(1)} > 0, s^{(2)} > 0)$. We construct SCRs for $V(\pi, s)$ over the domain $s \in [-1, 1]^2$. Similar to that in Scenario 1, we simulate 10000 independent trajectories, each initialized at a point in the grid $\{s : s^{(1)} = -1 + 2i/29, s^{(2)} = -1 + 2j/29, \ \text{for } 1 \leq i, j \leq 30\}$, to approximate the true value function $V(\pi, s), s \in [-1, 1]^2$. We construct SCRs using tensor products of Legendre and spline basis functions, respectively. The results are summarized in Table 1. Moreover, we conducted additional simulations employing SAVE with the Sidak correction (Abdi et al. 2007), and the results are summarized in Table S.2 in the supplement.

Furthermore, we modified the state transition rule to assess the performance of our method under high noise and non-Gaussian errors. Specifically, we set $z_{0,t}$ to be an i.i.d. two-dimensional $t(8)$ random

variable, while keeping all other components unchanged. The results are presented in Table S.3, where the coverage and length remain robust.

In addition to the comparison with SAVE, we also evaluated our method against the importance sampling approach (Jiang & Li 2016, Hanna et al. 2017) based on Scenario 2. The detailed experimental settings and results are provided in Section B in the supplement.

Table 1: Results for Scenario 2. Format: ECP(AL).

| $n$ | $T$ | Legendre | | Spline | |
|---|---|---|---|---|---|
| | | SCR | SAVE | SCR | SAVE |
| 30 | 50 | 0.926 (8.472) | 0.982 (9.445) | 0.936 (9.452) | 0.978 (9.793) |
| 50 | 30 | 0.946 (9.553) | 0.970 (10.492) | 0.922 (11.101) | 0.942 (11.131) |
| 40 | 50 | 0.924 (7.225) | 0.976 (8.193) | 0.938 (8.087) | 0.976 (8.606) |
| 50 | 40 | 0.944 (7.138) | 0.990 (8.136) | 0.930 (7.247) | 0.984 (7.753) |
| 50 | 50 | 0.942 (7.299) | 0.978 (8.249) | 0.930 (8.156) | 0.966 (8.630) |
| 50 | 150 | 0.952 (6.985) | 0.966 (7.402) | 0.934 (5.771) | 0.978 (6.080) |
| 50 | 200 | 0.944 (5.978) | 0.978 (6.436) | 0.950 (7.733) | 0.960 (7.418) |
| 50 | 250 | 0.942 (5.957) | 0.968 (6.234) | 0.930 (5.177) | 0.960 (5.446) |
| 200 | 70 | 0.934 (4.924) | 0.988 (5.370) | 0.926 (4.766) | 0.968 (5.045) |
| 250 | 70 | 0.936 (4.374) | 0.986 (4.800) | 0.906 (4.245) | 0.968 (4.521) |
| 300 | 70 | 0.934 (4.430) | 0.974 (4.737) | 0.936 (5.029) | 0.978 (5.031) |

Table 1 provides several key insights. First, it illustrates the theoretical claim that our method is primarily governed by the product $N = nT$. In addition, it indicates that both spline and Legendre bases lead to similar results, with the SCRs constructed using spline bases being slightly wider. This suggests some robustness of our method to the choice of basis functions, which is appealing for practical applications.

**Remark 4.1.** *Note that the empirical coverage probability (ECP) is the mean of binary outcomes. Therefore, we can derive the confidence interval for it. Specifically, the 95% confidence interval for ECP is given by $[p - 1.96\sqrt{p(1-p)/500},\ p + 1.96\sqrt{p(1-p)/500}]$, where $p$ denotes the empirical coverage.*

### 4.2 Real data application

In this section, we apply our method to the OhioT1DM dataset[3], which contains records of continuous glucose monitoring (CGM), insulin administration, and self-reported life events for six individuals diagnosed with type 1 diabetes. The data is partitioned into consecutive three-hour intervals and has a three-dimensional state variable $S_{i,t}$ for each patient $i$ at time step $t$. Due to the space limitation, we leave the specific construction in the supplement. The action $A_{i,t}$ is constructed as a binary variable. $A_{i,t} = 1$ if the cumulative insulin administered during the interval exceeds one unit; otherwise $A_{i,t} = 0$. The discount factor is set as $\gamma = 0.5$ to weight future outcome. The reward, $R_{i,t}$, is derived from the Index of Glycemic Control (IGC), a piecewise function that penalizes both hypoglycemia and hyperglycemia while assigning zero cost to glucose values within a clinically optimal range, i.e.,

$$R_{i,t} = -\left(80 - S_{i,t+1}^{(1)}\right)^2 I_{\{S_{i,t+1}^{(1)} < 80\}}/30 - \left(S_{i,t+1}^{(1)} - 140\right)^{1.35} I_{\{S_{i,t+1}^{(1)} \geq 140\}}/30.$$

The downloaded dataset has been separated as training group and testing group. Our objective is to conduct the simultaneous OPE on the testing group under the target policies obtained from the training group. In specific, we evaluate two kinds of target policies on the testing group. The first is an optimal policy $\pi^{opt}$ obtained by the double fitted Q-iteration algorithm ((Härdle & Song 2010)) in the training group; implementation details are provided in Section E of the supplementary material. The second is the behavior policy $b$ obtained by the random forest from the training data. We then estimate value functions $\hat{V}(\pi^{opt}, S_0)$ and $\hat{V}(b, S_0)$ on the testing set by (3.8). SCRs for $\hat{V}(\pi^{opt}, S_0)$ and $\hat{V}(b, S_0)$ are constructed for all states in the test set by Algorithm 1.

The results show that $\hat{V}(\pi^{opt}, S_0)$ exceeds $\hat{V}(b, S_0)$ by an average of 2.61, and improvements are observed in 87.1% of the initial states. To characterize uncertainty, we examine the proportion of states under which the SCRs do not cover 0 (i.e., the average CGM blood glucose level is not within

---

[3]https://www.kaggle.com/datasets/ryanmouton/ohiot1dm

the normal range) for both target policies. Owing to the uniform property of the SCR, the proportion of states for which the SCRs do not cover zero reflects the fraction of patients who remain in a significantly poor condition under the target policy. The results show that, at the 5% significance level, for policy $b$, the value function $\hat{V}(b, S_0)$ is significantly less than 0 in 90.7% of the states, whereas for policy $\hat{V}(\pi^{opt}, S_0)$, this proportion is 23.3%. We visualize the SCRs where the upper bound of 95% SCR is below than 0, sorted by the value estimates, in Figure 2. In terms of the average length, for $\hat{V}(\pi^{opt}, S_0)$, our method yields an averaged length of 27.0, while SAVE with Bonferroni correction produces an AL of 32.4, which is 20% larger than ours. Moreover, for $\hat{V}(b, S_0)$, our method yields an average length of 7.02, compared to 7.58 for SAVE (approximately 8% longer). These findings suggest that, in the medical context, applying reinforcement learning algorithms alongside simultaneous inference could improve health outcomes for patients.

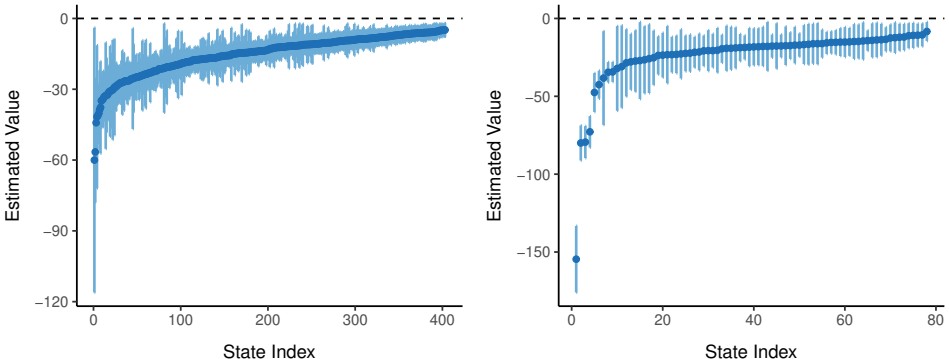

Figure 2: Left: Visualization of values where $\hat{V}(b, S_0)$ is sufficiently negative (the upper bound of 95% SCR is below 0). Right: Visualization of values where $\hat{V}(\pi^{opt}, S_0)$ is sufficiently negative (the upper bound of 95% SCR is below 0).

## 5    Conclusion and future work

In this work, we present a novel simultaneous statistical inference framework for off-policy evaluation, proving that our SCRs are asymptotically correct via convex Gaussian approximation. The SCRs have widths exceeding pointwise confidence intervals by only a logarithmic factor. This establishes near-optimal efficiency while achieving uniform coverage. The method's validity and efficiency are demonstrated both theoretically and empirically.

The current results are limited to offline settings. Extending this framework to online RL represents a natural next research direction. Additionally, the simultaneous inference framework shows potential for extension to more general Q-learning estimation in RL, including robust value estimation (e.g., Panaganti et al. (2022), Cayci & Eryilmaz (2023)).

## Acknowledgments and disclosure of funding

This work was supported by the High Performance Computing Center, Tsinghua University. Weichi Wu, the corresponding author, is supported by the NSFC No.12271287.

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

# Supplementary materials for "Simultaneous Statistical Inference for Off-Policy Evaluation in Reinforcement Learning"

**Tianpai Luo**[*]  **Xinyuan Fan**[*]  **Weichi Wu**[†]
Department of Statistics and Data Science
Tsinghua University
Beijing 100084, China
{ltp21, fxy22}@mails.tsinghua.edu.cn, wuweichi@tsinghua.edu.cn

The supplementary materials are organized as follows. In Section A, we discuss the key challenges in policy evaluation for offline reinforcement learning and describe methods for assessing dataset quality. In Section B, we report the additional simulation results described in the main article. In Section C, we detail the construction of the state space in the real data example. In Section D, we present the cross-validation method for selecting the number of basis functions. In Section E, we provide the double fitted Q-learning algorithm. In Section F, we introduce commonly used basis functions and verify the related conditions stated in the main paper. Section G contains the proofs of the theorems, along with the relevant lemmas.

Notations in this supplement are summarized as follows. For a vector $\mathbf{v} =: (v_1, v_2, \ldots, v_p) \in \mathbb{R}^p$, let $|\mathbf{v}| = \left(\sum_{i=1}^p v_i^2\right)^{1/2}$. For a random vector $\mathbf{V}$ and probability measure P, denote $\|\mathbf{V}\|_{\mathrm{P},q} =: \left[\mathbb{E}_{\mathrm{P}}\left(|\mathbf{V}|^q\right)\right]^{1/q}$, $q > 0$ where $\mathbb{E}_{\mathrm{P}}(\cdot)$ is the expectation with respect to probability P. For simplicity, we shall use $\mathbb{E}(\cdot)$, $\|\cdot\|_q$, $\|\cdot\|$ instead of $\mathbb{E}_{\mathrm{P}}(\cdot)$, $\|\cdot\|_{\mathrm{P},q}$, $\|\cdot\|_{\mathrm{P},2}$, respectively if no confusion arises. For a matrix $\mathbf{A}$, the determinant of a matrix $\mathbf{A}$ is denoted as $\det(\mathbf{A})$. If the matrix $\mathbf{A}$ is real and symmetric, we use $\lambda_{min}(\mathbf{A})$ ($\lambda_{max}(\mathbf{A})$) to denote the smallest (largest) eigenvalue of $\mathbf{A}$. For any two positive real sequences $a_n$ and $b_n$, write $a_n \asymp b_n$ if there exists $0 < c < C < \infty$ such that $c \leq \liminf_{n\to\infty} a_n/b_n \leq \limsup_{n\to\infty} a_n/b_n \leq C$. We write $a_n \lesssim b_n$ ($a_n \gtrsim b_n$) to mean that there exists a universal constant $C > 0$ such that $a_n \leq C b_n$ ($C a_n \geq b_n$) for all $n$.

## A  Offline Reinforcement Learning: Evaluation Challenges and Dataset Quality

Unlike online environments (e.g., Go or MiniGrid), collecting data through online interaction in many real-world applications, such as healthcare or autonomous driving, can be costly or even hazardous. This limitation hinders the widespread adoption of traditional online RL methods. As an alternative, offline RL leverages large historical datasets, and it is particularly suited to situations where online interaction is infeasible but existing data is available for learning.

Nevertheless, offline RL introduces unique challenges compared to online RL. From the learning perspective, key difficulties include distributional shift, where the behavior policy used to collect the dataset may differ from the target policy being learned, potentially leading to poor performance or overfitting; and sample inefficiency, since learning relies entirely on a static dataset, preventing online exploration. From the evaluation perspective, estimating policy performance without online deployment necessitates off-policy evaluation (OPE), which is the focus of our work. A central challenge in OPE is the gap between estimated off-policy performance and real-world outcomes. Recent methods, such as pointwise confidence intervals, aim to quantify this gap probabilistically.

---

[*]Equal contribution.
[†]Corresponding author.

39th Conference on Neural Information Processing Systems (NeurIPS 2025).

Our work extends these tools from pointwise to global inference, providing more reliable guarantees for decision-making.

The quality of an offline dataset critically affects learning and evaluation. Important factors include state-space coverage, reward signal diversity, action variety, and compatibility between the data distribution and the RL algorithm. As noted in Levine et al. (2020), formalizing a non-trivial sufficiency condition for dataset quality remains an open problem. In our framework, these considerations are captured through Assumptions (A1)–(A3), under which the dataset's quality can be roughly quantified by its sample size $N$. Our empirical results demonstrate reliable performance of our large-sample theory with dataset sizes around $N = 2000$, outperforming modified classical methods such as SAVE when sample sizes are larger. This scale is modest compared to typical offline RL applications: for instance, the HiRID Rodemund et al. (2023) and MIMIC-III Johnson et al. (2016) datasets contain extensive ICU records collected over multiple years, and datasets like D4RL (Datasets for Deep Data-Driven Reinforcement Learning) provide large-scale data for computer science applications. These examples highlight that our approach can be reliably applied across a wide range of real-world scenarios.

## B  Additional simulation results

### B.1  Sensitivity analysis for scenario 1

We perform the sensitivity analysis for scenario 1 by taking $(n, T) = (25, 50), (50, 50)$ as two illustrative examples and examining the results by varying the specification of $K$ over a relatively wide range. The results are presented in Table S.1. It can be seen that our method is not sensitive to the choice of the number of basis functions $K$.

Table S.1: Sensitivity analysis under Scenario 1 for different values of $K$.

| $n$ | $T$ | $K$ | ECP | AL |
|---|---|---|---|---|
| 25 | 50 | 10 | 0.92 | 0.121 |
| | | 11 | 0.93 | 0.127 |
| | | 12 | 0.94 | 0.133 |
| | | 13 | 0.92 | 0.137 |
| 50 | 50 | 20 | 0.93 | 0.120 |
| | | 22 | 0.94 | 0.127 |
| | | 24 | 0.96 | 0.135 |
| | | 26 | 0.95 | 0.143 |

### B.2  SAVE with the Sidak correction for Secnario 2

The results for SAVE with the Sidak correction in Scenario 2 are summarized in Table S.2. Results with high noise and non-Gaussian errors are reported in Table S.3.

Table S.2: Results for Scenario 2 using SAVE with Sidak correction. Format: ECP(AL).

| $n$ | $T$ | Legendre SAVE (Sidak) | Spline SAVE (Sidak) |
|---|---|---|---|
| 30 | 50 | 0.980 (9.431) | 0.976 (9.778) |
| 50 | 30 | 0.968 (10.476) | 0.942 (11.115) |
| 40 | 50 | 0.976 (8.181) | 0.974 (8.593) |
| 50 | 40 | 0.990 (8.124) | 0.984 (7.741) |
| 50 | 50 | 0.978 (8.237) | 0.966 (8.617) |
| 50 | 150 | 0.974 (7.393) | 0.972 (6.075) |
| 50 | 200 | 0.974 (6.435) | 0.966 (7.385) |
| 50 | 250 | 0.952 (6.223) | 0.980 (5.431) |
| 200 | 70 | 0.972 (5.360) | 0.978 (5.050) |
| 250 | 70 | 0.972 (4.804) | 0.958 (4.503) |
| 300 | 70 | 0.986 (4.726) | 0.972 (5.020) |

Table S.3: Results for Scenario 2 with $t(8)$ noises. Format: ECP(AL).

| $n$ | $T$ | Legendre SCR | Legendre SAVE (Bonferroni) | Legendre SAVE (Sidak) | Spline SCR | Spline SAVE (Bonferroni) | Spline SAVE (Sidak) |
|---|---|---|---|---|---|---|---|
| 30 | 50 | 0.940 (4.706) | 0.934 (4.431) | 0.980 (4.424) | 0.936 (3.716) | 0.934 (3.967) | 0.986 (3.961) |
| 50 | 30 | 0.944 (2.721) | 0.932 (3.316) | 0.990 (3.311) | 0.934 (3.694) | 0.932 (3.959) | 0.988 (3.953) |
| 40 | 50 | 0.946 (3.349) | 0.934 (3.658) | 0.992 (3.652) | 0.952 (3.632) | 0.950 (3.755) | 0.990 (3.749) |
| 50 | 40 | 0.932 (2.876) | 0.928 (3.279) | 0.990 (3.274) | 0.944 (3.104) | 0.942 (3.393) | 0.988 (3.388) |
| 50 | 50 | 0.934 (3.488) | 0.926 (3.448) | 0.986 (3.443) | 0.934 (2.776) | 0.932 (3.048) | 0.994 (3.044) |
| 50 | 150 | 0.948 (2.333) | 0.944 (2.498) | 0.988 (2.494) | 0.948 (2.415) | 0.944 (2.524) | 0.988 (2.520) |
| 50 | 200 | 0.942 (2.012) | 0.946 (2.168) | 0.984 (2.165) | 0.936 (2.078) | 0.934 (2.184) | 0.988 (2.181) |
| 50 | 250 | 0.932 (1.962) | 0.934 (2.111) | 0.994 (2.108) | 0.950 (2.330) | 0.944 (2.332) | 0.986 (2.329) |
| 200 | 70 | 0.946 (1.970) | 0.996 (2.024) | 0.996 (2.021) | 0.944 (2.176) | 0.944 (2.196) | 0.980 (2.192) |
| 250 | 70 | 0.948 (1.644) | 0.980 (1.784) | 0.980 (1.780) | 0.944 (1.943) | 0.944 (1.968) | 0.982 (1.965) |
| 300 | 70 | 0.930 (1.495) | 0.986 (1.625) | 0.986 (1.623) | 0.948 (1.770) | 0.944 (1.800) | 0.980 (1.797) |

### B.3 Comparison with the importance sampling method for Scenario 2

We evaluated our method against the importance sampling (IS) approach (Jiang & Li 2016, Hanna et al. 2017) under Scenario 2. Specifically, we set $S_0 = (-2+0.4i, -2+0.4j)^\top$ where $0 \le i, j \le 10$. For each combination of $i$ and $j$, we generated $n_0 = 100$ trajectories, each of length 10, while keeping all other settings unchanged. For bootstrapping IS, we employed the algorithm from Hanna et al. (2017), which provides confidence intervals for each $V(\pi, S_0)$. The Bonferroni correction was then applied to obtain the simultaneous confidence bands (SCB).

It is worth noting that IS approach estimates the value function by directly reweighting trajectories, whereas in Scenario 2, the target policy $\pi(1|s) = I(s^{(1)} > 0, s^{(2)} > 0)$ is discontinuous in $s$. Moreover, the choice of target policy frequently results in weights of zero, reducing the effective sample size and causing the IS estimates to be dominated by a small subset of samples. This leads to biased estimates and, consequently, degraded performance. We mention that Hanna et al. (2018) also highlighted the same issue in their Section 7. Our method is not impacted by this problem, further showcasing its practical applicability.

From the results, under this setting, the IS method achieves an empirical coverage probability (ECP) of 0.638, with an average length (AL) of 5.300. This is below the nominal level of 0.95. Increasing the sample size can improve the coverage of the IS method, however, the associated bootstrap procedure becomes computationally expensive. In contrast, our method performs well even with a smaller sample size ($n_0 = 10$), achieving an empirical coverage probability (ECP) of 0.954 and an average length (AL) of 4.575.

## C Specific construction for states in the real data example

In the real data example in the main article, we construct a three-dimensional state variable $S_{i,t}$ for each patient $i$ at time step $t$. Specifically, $S_{i,t}^{(1)}$ represents the average CGM blood glucose level over the preceding three-hour interval. $S_{i,t}^{(2)}$ is a decayed sum of carbohydrate intake within the same period, where each meal's carbohydrate estimate is discounted according to its temporal distance from the current interval. Specifically, if meals are recorded at times $t_1, t_2, \ldots, t_N \in [t-1, t)$ with corresponding carbohydrate estimates $CE_1, CE_2, \ldots, CE_N$, then $S_{i,t}^{(2)} = \sum_{j=1}^{N} CE_j \cdot 0.5^{36(t_j - t + 1)}$. $S_{i,t}^{(3)}$ denotes the average basal rate over the same three-hour window, capturing the background level.

## D Cross-validation for choosing the number of basis functions

The method of cross-validation is widely used in machine learning and sieve methods (see, for example, Van Der Laan & Dudoit (2003), Hansen (2014), Bates et al. (2024)). Based on the key equation (3.5) in the main article, we adopt the following 5-fold cross-validation approach, as described in Algorithm S.1.

---

**Algorithm S.1** 5-Fold Cross-Validation

---

1: **Input:** Observed data $\mathcal{D} = \{(R_{i,t}, A_{i,t}, S_{i,t}, S_{i,t+1})\}_{0 \leq t \leq T_i, \, 1 \leq i \leq n}$; candidate set of choices $K_{\text{can}} = \{k_1, \ldots, k_l\}$.
2: Randomly partition $\mathcal{D}$ into 5 approximately equal-sized folds: $\mathcal{D}_1, \mathcal{D}_2, \mathcal{D}_3, \mathcal{D}_4, \mathcal{D}_5$.
3: **for** $j = 1$ to 5 **do**
4:     Set the $j$-th fold as validation set: $\mathcal{D}_{\text{val}} \leftarrow \mathcal{D}_j$.
5:     Set the remaining 4 folds as training set: $\mathcal{D}_{\text{train}} \leftarrow \bigcup_{i \neq j} \mathcal{D}_i$.
6:     **for** $k = 1$ to $l$ **do**
7:         Obtain $\hat{\beta}_\pi^{(j,k)}$ based on $\mathcal{D}_{\text{train}}$ using equation (3.6) with $k_k$ basis functions.
8:         Let

$$CV(j,k) = \sum_{\mathcal{D}_{\text{val}}} \left\{ \left( R_{i,t} + \gamma \sum_{a \in \mathcal{A}} \Phi(S_{i,t+1})^\top \hat{\beta}_{\pi,a}^{(j,k)} \pi(a|S_{i,t+1}) \right. \right.$$
$$\left. \left. - \Phi(S_{i,t})^\top \hat{\beta}_{\pi,A_{i,t}}^{(j,k)} \right) \Phi(S_{i,t}) \right\}^2 .$$

9:     **end for**
10: **end for**
11: Let $k^* = \arg\min_{k=1,\cdots,l} \sum_{j=1}^5 CV(j,k)$.
12: **Output:** Select $k_{k^*}$ as the number of basis functions.

## E  Double-fitted Q-iteration algorithm

The double-fitted Q-iteration algorithm (Hasselt 2010) is presented in Algorithm S.2. The Q-function $Q(\cdot, \cdot; \theta)$ can be specified using any model indexed by $\theta$, and we use a linear model with basis functions to approximate $Q$.

---

**Algorithm S.2** Double Fitted Q-Iteration Algorithm

---

1: **Input:** Observed data $\{(R_{i,t}, A_{i,t}, S_{i,t}, S_{i,t+1})\}_{0 \leq t \leq T_i, \, 1 \leq i \leq n}$; initialize parameters $\hat{\theta}_A, \hat{\theta}_B$.
2: **repeat**
3:     **Step 1:** For all $i, t$, compute target values:

$$\hat{Q}_{i,t}^A = R_{i,t} + \gamma Q \left( S_{i,t+1}, \arg\max_{a'} Q(S_{i,t+1}, a'; \hat{\theta}_A), \hat{\theta}_B \right)$$
$$\hat{Q}_{i,t}^B = R_{i,t} + \gamma Q \left( S_{i,t+1}, \arg\max_{a'} Q(S_{i,t+1}, a'; \hat{\theta}_B), \hat{\theta}_A \right)$$

4:     **Step 2:** Update parameters by minimizing squared errors:

$$\hat{\theta}_A \leftarrow \arg\min_{\theta_A} \sum_i \sum_t \left\| Q(X_{i,t}, A_{i,t}; \theta_A) - \hat{Q}_{i,t}^A \right\|^2$$
$$\hat{\theta}_B \leftarrow \arg\min_{\theta_B} \sum_i \sum_t \left\| Q(X_{i,t}, A_{i,t}; \theta_B) - \hat{Q}_{i,t}^B \right\|^2$$

5: **until** convergence
6: **Output:** Learned parameter $\hat{\theta}_A$.

---

## F  Sieve method

In this section, we introduce commonly used sieve basis and verify the related conditions in the main paper. We list several commonly used sieve basis as follows, which can be used in our simultaneous inference framework.

**Example F.1** (Legendre). *Define Legendre polynomials*

$$P_j(x) = \frac{1}{2^j \, j!} \frac{\mathrm{d}^j}{\mathrm{d}x^j} (x^2 - 1)^j, \quad x \in [-1, 1].$$

*Then continuous function $f(x)$ on $[-1, 1]$ can be written as*

$$f(x) = \sum_{j=0}^\infty a_j P_j(x), \quad a_j = (j + \frac{1}{2}) \int_{-1}^1 P_j(x) f(x) \mathrm{d}x.$$

**Example F.2** (Fourier). *Consider real-valued function $f(x) \in L_2[-1, 1]$ i.e. $\int_{-1}^{1} f(x)\mathrm{d}x < \infty$. By Fourier transformation, $f(x)$ can be written as*

$$f(x) = \sum_{j=-\infty}^{\infty} a_j \phi_j(x), \quad a_j = \int_{-1}^{1} \phi_j(x) f(x)\mathrm{d}x$$

*where $\{\phi_j(x)\}_{j=-\infty}^{\infty} = \{(\cos(j\pi x) + i\sin(j\pi x))/\sqrt{2}\}_{j=-\infty}^{\infty}$ forms an orthonormal basis for $L_2[-1, 1]$.*

**Example F.3** (Harr wavelet). *The Haar sequence was proposed in 1909 by Haar (1910). Haar used these functions to give an example of an orthonormal system for the space of square-integrable functions. For every pair $n, k$ of integers in $\mathbb{Z}$, the Haar function $h_{n,k}$ is defined on the real line $\mathbb{R}$ by the formula*

$$h_{n,k}(t) = 2^{n/2} h\left(2^n t - k\right),$$

*where $h(t)$ is the Harr wavelet's mother wavelet function*

$$h(t) = \begin{cases} 1 & 0 \leq t < \frac{1}{2} \\ -1 & \frac{1}{2} \leq t < 1 \\ 0 & otherwise \end{cases}.$$

*The Haar system on the real line is the set of functions*

$$\{h_{n,k}(t) : n \in \mathbb{Z}, k \in \mathbb{Z}\},$$

*which is an orthonormal basis.*

**Example F.4** (Daubechies wavelet). *For $N \in \mathbb{N}$, a Daubechies mother wavelet of class Daubechies-$N$ is a function $\phi \in L_2(\mathbb{R})$ defined by*

$$\phi(x) := \sqrt{2} \sum_{k=1}^{2N-1} (-1)^k h_{2N-1-k} \varphi(2x - k),$$

*where $h_0, h_1, \cdots, h_{2N-1} \in \mathbb{R}$ are constant and satisfy $\sum_{k=0}^{N-1} h_{2k} = \frac{1}{\sqrt{2}} = \sum_{k=0}^{N-1} h_{2k+1}$, as well as, for $l = 0, 1, \cdots, N - 1$,*

$$\sum_{k=2l}^{2N-1+2l} h_k h_{k-2l} = \begin{cases} 1, & l = 0 \\ 0, & l \neq 0 \end{cases}$$

*The $\varphi(x)$ is the scaling wavelet function supported on $[0, 2N - 1)$ and satisfies the recursion equation $\varphi(x) = \sqrt{2} \sum_{k=0}^{2N-1} h_k \varphi(2x - k)$, as well as the normalization $\int_{\mathbb{R}} \varphi(x)dx = 1$, $\int_{\mathbb{R}} \varphi(2x - k)\varphi(2x - l)dx = 0, k \neq l$. As listed in Daubechies (1992), the filter coefficients $h_0, \ldots, h_{2N-1}$ can be efficiently computed. The order $N$ decides the support $[0, 2N - 1)$ and provides the regularity condition*

$$\int_{\mathbb{R}} x^j \phi(x)\mathrm{d}x = 0, j = 0, \cdots, N.$$

*The Harr wavelet as introduced above can be regarded as a special Daubechies wavelet with $N = 1$. In our simulations and data analysis, we employ Daubechies wavelet with a sufficiently high order $N$ to construct a sequence of orthogonal sieve basis as proposed in Daubechies (1988). For a given $J_n$ and $J_0$, we consider the following periodized wavelets on $[0, 1]$*

$$\left\{\varphi_{J_0 k}(x), 0 \leq k \leq 2^{J_0} - 1; \phi_{jk}(x), J_0 \leq j \leq J_n - 1, 0 \leq k \leq 2^j - 1\right\}, \text{ where}$$

$$\varphi_{J_0 k}(x) = 2^{J_0/2} \sum_{l \in \mathbb{Z}} \varphi\left(2^{J_0} x + 2^{J_0} l - k\right), \phi_{jk}(x) = 2^{j/2} \sum_{l \in \mathbb{Z}} \psi\left(2^j x + 2^j l - k\right)$$

*or equivalently, by Yves (1989),*

$$\left\{\varphi_{J_n k}(x), 0 \leq k \leq 2^{J_n - 1}\right\}.$$

*The $2_n^J$ equals to our basis number $K$. Additionally, we refer to Chen (2007) for a more general example of orthogonal wavelets.*

## F.1 Sieve approximation

For the approximation (3.4), we show that the sieve method can approximate any function in the Hölder space with smoothness $p$. Given $d$-tuple $\alpha = (\alpha_1, \ldots, \alpha_d)$ of nonnegative integers and $[\alpha] = \alpha_1 + \cdots + \alpha_d$, the Hölder space with smoothness $p$, $\Lambda_C^p(\mathcal{S})$, is defined as

$$\Lambda_C^p(\mathcal{S}) =: \left\{ h \in \mathcal{C}^m(\mathcal{S}) : \sup_{[\alpha] \leq m} \sup_{s \in \mathcal{S}} |D^\alpha h(s)| \leq C, \sup_{[\alpha] = m} \sup_{x, y \in \mathcal{S}, x \neq y} \frac{|D^\alpha h(x) - D^\alpha h(y)|}{|x - y|^\gamma} \leq C \right\}, \tag{S.1}$$

where $C > 0$ is a constant, $p = m + \gamma, \gamma \in (0, 1]$, $\mathcal{C}^m(\mathcal{S})$ is the class of $m$-times continuously differentiable real-valued functions on $\mathcal{S}$, and the differential operator

$$D^\alpha = \frac{\partial^{[\alpha]}}{\partial x_1^{\alpha_1} \cdots \partial x_d^{\alpha_d}}.$$

For function $Q(\pi; \cdot, a) \in \Lambda_C^p(\mathcal{S})$, $\sup_{s \in \mathcal{S}, a \in \mathcal{A}} |Q(\pi; s, a) - \Phi(s)^\top \beta_{\pi, a}^*| = O(K^{-p/d})$ if $\Phi(s)$ is the tensor product of sieve bases such as B-splines, Legendre polynomials, orthogonal wavelets, or Fourier series if it is periodic; see Section 2.3.1 in Chen (2007) or Timan (2014),Yves (1989),Chen (2007). As discussed in Shi, Wan, Chernozhukov & Song (2021), there exists some transition density function $q$ such that $\mathcal{P}(\mathrm{d}s', a) = q(s'|s, a)\mathrm{d}s$ if the transition kernel $\mathcal{P}(\cdot|s, a)$ is absolutely continuous with respect to the Lebesgue measure. The following Lemma shows that $Q(\pi; \cdot, a) \in \Lambda_C^p(\mathcal{S})$ if $q(s'|\cdot, a)$ and reward $r(s, a)$ follow certain mild conditions.

**Lemma F.1** (Lemma 1 in Shi, Zhang, Lu & Song (2021))**.** *If there exist some $p, C > 0$ such that $r(\cdot, a), q(s'|\cdot, a) \in \Lambda_C^p(\mathcal{S})$ for any $a \in \mathcal{A}, s' \in \mathcal{S}$, then there exists constant $C' > 0$ such that $Q(\pi; ;a) \in \Lambda_{C'}^p(\mathcal{S})$ for any policy $\pi$ and $a \in \mathcal{A}$.*

## F.2 Geometric properties of sieve space

In this section, we verify the condition (3.15) in Proposition 3.2. Condition (3.15) are simplified requirement on the sieve basis which will yield a polynomial rate $N^c$ ($c \geq 0$) for the geometric quantities, including volume, curvature, and boundary of the manifold $\{\Phi(s)/|\Phi(s)| : s \in \mathcal{S}\}$. For simplicity, we only verify $\int_\mathcal{S} \lambda_{\min}^{d/2}(\mathbf{M}^\top \mathbf{M})\mathrm{d}s \gtrsim N^{\underline{c}}$ in condition (3.15). We refer to Assumption 4 of Chen & Christensen (2015) and Example 1-2 in Quan & Lin (2024) for the rest polynomial rate conditions in (3.15). Define $\xi_{K,N} =: \sup_{s \in \mathcal{S}} |\Phi(s)|$ and $\Delta_{K,N} =: \sup_{s \in \mathcal{S}} |\nabla\Phi(s)|$. Then there exists $\bar{\omega}, \omega_0, \omega_1, \omega_1' \geq 0$ s.t. $\xi_{K,N} \lesssim N^{\omega_1}$, and $\Delta_{K,N} \lesssim N^{\omega_1'}$ and $N^{\bar{\omega}} \ll K \ll N^{\omega_0}$.

**Lemma F.2** (Lemma E.1 in Shi, Zhang, Lu & Song (2021))**.** *There exists some constant $c^* \geq 1$ such that*

$$(c^*)^{-1} \leq \lambda_{\min} \left\{ \int_{s \in \mathcal{S}} \Phi(s)\Phi^\top(s)\mathrm{d}s \right\} \leq \lambda_{\max} \left\{ \int_{s \in \mathcal{S}} \Phi(s)\Phi^\top(s)\mathrm{d}s \right\} \leq c^*$$

*and $\xi_{K,N} \leq c^*\sqrt{K}$.*

We verify condition (3.15) using trigonometric basis functions as a representative example. A similar procedure applies to other types of basis functions.

Suppose that $d = 1, \mathcal{S} = [-\pi, \pi]$, the number of basis functions is $K = 2\tilde{K} + 1, \tilde{K} \geq 1$, and $\Phi(s) = (1, \sin(s), \cos(s), \cdots, \sin(\tilde{K}s), \cos(\tilde{K}s))^\top$. Then $|\Phi(s)| = \sqrt{\tilde{K} + 1}, M(s) = (0, \cos(s), -\sin(s), \cdots, \tilde{K}\cos(\tilde{K}s), -\tilde{K}\sin(\tilde{K}s))^\top/\sqrt{\tilde{K} + 1}$, and $M^\top M = \sum_{i=1}^{\tilde{K}} i^2/(\tilde{K}+1) \gtrsim \tilde{K}^2$. As a result, we have $\int_\mathcal{S} \lambda_{min}(\mathbf{M}^\top \mathbf{M})\mathrm{d}s \gtrsim K \gtrsim N^{\bar{\omega}}$.

Now consider the case where $d = 2$ and $\mathcal{S} = [-\pi, \pi]^2$. Suppose that the number of basis functions is $K = (2\tilde{K} + 1)^2$. Then $\Phi(s) = \phi(s_1) \otimes \phi(s_2)$ where $\phi(s) = (1, \sin(s), \cos(s), \cdots, \sin(\tilde{K}s), \cos(\tilde{K}s))^\top$. $M_1(s) = \psi(s_1) \otimes \phi(s_2)/(\tilde{K} + 1), M_2(s) = \phi(s_1) \otimes \psi(s_2)/(\tilde{K} + 1)$ where $\psi(s) = (0, \cos(s), -\sin(s), \cdots, \tilde{K}\cos(\tilde{K}s), -\tilde{K}\sin(\tilde{K}s))^\top$.

$$M^\top M = \frac{1}{(\tilde{K} + 1)^2} \begin{pmatrix} \psi(s_1)^\top \psi(s_1)\phi(s_2)^\top \phi(s_2) & \psi(s_1)^\top \phi(s_1)\phi(s_2)^\top \psi(s_2) \\ \psi(s_2)^\top \phi(s_2)\phi(s_1)^\top \psi(s_1) & \phi(s_1)^\top \phi(s_1)\psi(s_2)^\top \psi(s_2) \end{pmatrix}$$

$$= \frac{1}{(\tilde{K} + 1)^2}\frac{1}{6}\tilde{K}(\tilde{K} + 1)^2(2\tilde{K} + 1) \begin{pmatrix} 1 & 0 \\ 0 & 1 \end{pmatrix} = \frac{1}{6}\tilde{K}(2\tilde{K} + 1) \begin{pmatrix} 1 & 0 \\ 0 & 1 \end{pmatrix}.$$

Then we have $\int_{\mathcal{S}} \lambda_{min}(\mathbf{M}^\top \mathbf{M}) \mathrm{d}s \gtrsim K \gtrsim N^{\bar{\omega}}$.

## G  Technical proofs

### G.1  Dependence measure and geometric ergodicity

To measure the dependence in the Markov chain, we introduce the concept of physical dependence measure in Wu (2005). For a pair of jointly distributed random variables $(X, Y)$, let $F_{XY}(x, y) = \mathbb{P}(X \leq x, Y \leq y), x, y \in \mathbb{R}$, be the joint distribution function and $F_{Y|X}(y \mid x) = \mathbb{P}(Y \leq y \mid X = x)$ the conditional distribution function of $Y$ given $X = x$. For $u \in (0, 1)$, define the conditional quantile function $G(x, u) = \inf\{y \in \mathbb{R} : F_{Y|X}(y \mid x) \geq u\}$. Let $U$ be a uniform $(0, 1)$ distributed random variable and assume that $U$ and $X$ are independent. Then we can view $Y$ as the outcome of the bivariate function $Y =_d G(X, U)$ such that

$$(X, Y) =_d (X, G(X, U)). \tag{S.1}$$

For many standard constructions of stochastic processes (see e.g. Deák (1990), chapter 5), a stochastic process $\{X_t\}$ can be represented as

$$\begin{pmatrix} X_1 \\ X_2 \\ \dots \\ X_n \end{pmatrix} =_d \begin{pmatrix} X_1 \\ G_2(X_1, U_2) \\ \dots \\ G_n(X_{n-1}, U_n) \end{pmatrix} =_d \begin{pmatrix} H_1(U_1) \\ H_2(U_2) \\ \dots \\ H_n(U_n) \end{pmatrix} \tag{S.2}$$

where $H_1, \dots, H_n$ are measurable functions. The above representation can characterize many Markov sequences (see e.g. Rüschendorf & de Valk (1993)). If $\{X_t\}$ is a Markov chain, then the conditional quantile $G_t(X_{t-1}, U_t)$ can be viewed as a function of $X_{t-1}$. Wiener (1958) first considered this representation problem for stationary and ergodic processes. For a Markov chain $\{X_t\}$ with the form $X_i = G_i(X_{i-1}, U_i)$, Wu & Mielniczuk (2010) asserts that, there exists a copy $\widetilde{X}_i$ of $X_i$ such that $\left(\widetilde{X}_i\right)_{i \in \mathbb{Z}} =_d (X_i)_{i \in \mathbb{Z}}$ and $\widetilde{X}_i$ is expressed as $H_i(\dots, U_{i-1}, U_i)$, a function of iid random variables.

**Lemma G.1** (Theorem 4.1 in Wu & Mielniczuk (2010)). *Assume that $\{X_i\}$ satisfies the recursion*

$$X_i = G_i(X_{i-1}, U_i) =: F_i(X_{i-1}), i \in \mathbb{Z}, \tag{S.3}$$

*where $U_i$ are i.i.d. standard uniform random variables. Here $F_i$ are independent random maps $F_i(x) = G_i(x, U_i)$. Assume that for some $\alpha > 0$ we have*

$$\sup_{i \in \mathbb{Z}} L_i < 1, \text{ where } L_i = \sup_{x \neq y} \frac{\|G_i(x, U) - G_i(y, U)\|_\alpha}{|x - y|}$$

*and for some $x_0$,*

$$\sup_{i \in \mathbb{Z}} \|G_i(x_0, U)\|_\alpha < \infty.$$

*Then the backward iteration $F_i \circ F_{i-1} \circ F_{i-2} \dots$ converges almost surely and the limit forms a non-stationary Markov chain which is a solution to* (S.3).

Let $\varepsilon_i, i \in \mathbb{Z}$, be i.i.d. random variables and let $\mathcal{H}_i = (\dots, \zeta_{i-1}, \zeta_i)$. Based on Lemma G.1, we can consider the irreducible and aperiodic Markov chain $\{X_i\}$ as

$$X_i = H_i(\mathcal{H}_i), \quad i \in \mathbb{Z} \tag{S.4}$$

where $H_i$ are measurable functions. We view $\xi_i$ as the input and $X_i$ as the output of the system. If $H_i$ does not depend on $i$, i.e., $H_i \equiv H$, then the process $(X_i)$ is stationary. Then we introduce the following definition to measure the dependence of $\{X_i\}$ in (S.4): Let $\{\zeta_i'\}$ be an i.i.d. copy of $\{\zeta_i\}$, denote $\mathcal{H}_{i,k} = (\mathcal{H}_{i-k-1}, \zeta_{i-k}', \zeta_{i-k+1}, \dots, \zeta_i)$.

**Definition G.1** (physical dependence measure). *Assume that $\sup_i \|H_i(\mathcal{H}_i)\|_q < \infty$ for $q > 0$, then we can define the physical dependence measure of $\{X_i\}$ as*

$$\delta_H(k, q) =: \sup_{i \in \mathbb{Z}} \|H_i(\mathcal{H}_i) - H_i(\mathcal{H}_{i,k})\|_q, \tag{S.5}$$

*where $\mathcal{H}_{i,k} = (\mathcal{H}_{i-k-1}, \zeta_{i-k}', \zeta_{i-k+1}, \dots, \zeta_i)$.*

Note that the Lemma G.1 and Definition G.1 allow a nonstationary Markov chain; our theorem can actually be generalized to nonstationary cases, which can be a promising future work.

For our stationary state observation $\{S_{0,t}\}$, by Lemma G.1, our **geometric ergodicity** assumes $\{S_{0,t}\}$ is an irreducible and aperiodic Markov chain where there exists $\{\tilde{S}_{0,t}\} = S(\mathcal{H}_t)$ in the form of (S.4) such that $\{\tilde{S}_{0,t}\} =_d \{S_{0,t}\}$ and the physical dependence measure is geometrically decaying, i.e. $\delta_S(k,1) = O(\chi^k)$ for some constant $\chi \in (0,1)$. In fact, for contracting Markov chains (e.g., autoregressive models), this assumption generally holds. Notice that

$$\delta_S(k,q) \leq \sup_{s \in \mathcal{S}} |s|^{(q-1)/q} (\delta_S(k,1))^{1/q}, \tag{S.6}$$

$\delta_S(k,q) = O(\chi^k)$ holds for any given $q \in \mathbb{N}^+$ since state space $\mathcal{S}$ is bounded. Furthermore, denote $\Phi(S_{0,t}) = G(\mathcal{H}_t) = \Phi \circ S(\mathcal{H}_t)$, then the physical dependence measure

$$\delta_\Phi(k,q) =: \|G(\mathcal{H}_i) - G(\mathcal{H}_{i,k})\|_q = O(\Delta_{K,N}\delta_S(k,q)) = O(\Delta_{K,N}\chi^k).$$

Note that $|\Phi(S_{0,t})| \leq \sup_s |\Phi(s)| = \xi_{K,N}$. Using the fact $\min\{x,1\} \leq x^\alpha, x \geq 0$ for any given $\alpha \in (0,1)$, we can have $\delta_\Phi(k,q) = O(\xi_{K,N}\Delta_{K,N}^\alpha \chi^{\alpha k})$ for any given $\alpha \in (0,1)$.

## G.2   Proof of Theorem 3.1

We introduce the following lemmas before proving Theorem 3.1. In the proof of Theorem 3.1 and the following Lemmas, we will omit the subscript $\pi$ in $U_\pi(\cdot), u_\pi(\cdot), \Sigma_\pi, \hat{\Sigma}_\pi, \hat{\beta}_\pi, \beta_\pi^*, \omega_\pi$, etc, for brevity. For simplicity, we deduce our proof under the condition (3.4). In other words, we consider the Q-function $Q^*(\pi; s, a) = \Phi(s)^\top \beta_{\pi,a}^*$ instead of $Q(\pi; s, a)$. This can be achieved when the $Q$-function is smoothing enough as discussed in Section F.1. We denote the dimension of $\hat{\beta}$ as $p =: mK$ where $p \asymp K$ since $m$ is fixed. For the convex Gaussian approximation, we introduce a smoothed function

$$h_{A,\epsilon_1}(\omega) =: h\left(\frac{\inf_{\nu \in A} |\omega - \nu|}{\epsilon_1}\right), \tag{S.7}$$

where $\omega \in \mathbb{R}^p$, convex set $A \subset \mathbb{R}^p$, and

$$h(x) = \begin{cases} 1, & x < 0, \\ 1 - 2x^2, & 0 \leqslant x < \frac{1}{2}, \\ 2(1-x)^2, & \frac{1}{2} \leqslant x < 1, \\ 0, & x \geqslant 1. \end{cases} \tag{S.8}$$

To show the convex Gaussian approximations, we introduce the following Lemmas.

**Lemma G.2** (Lemma 5.3 in Fang & Koike (2024)). *For any $p$-dimensional random vector $\mathbf{W}$,*

$$\sup_{\mathbb{O} \in \mathfrak{D}} |P(\mathbf{W} \in \mathbb{O}) - P(\mathbf{Z} \in \mathbb{O})| \leq 4p^{1/4}\epsilon_1 + \sup_{\mathbb{O} \in \mathfrak{D}} |\mathbb{E}h_{\mathbb{O},\epsilon_1}(\mathbf{W}) - \mathbb{E}h_{\mathbb{O},\epsilon_1}(\mathbf{Z})|,$$

*where $\mathbf{Z}$ is a $p$-dimensional Gaussian random vector with invertible covariance matrix and $\mathfrak{D}$ is the collection of all the convex sets in $\mathbb{R}^p$.*

**Lemma G.3** (Theorem 2.1 in Fang (2016)). *Let $W = \sum_{i=1}^n X_i$ be a sum of $p$-dimensional random vectors such that $\mathbb{E}(X_i) = 0$ and $\mathrm{Cov}(W) = \Sigma$. Suppose $W$ can be decomposed as follows:*

*1. $\forall i \in [n], \exists i \in N_i \subset [n]$, such that $W - X_{N_i}$ is independent of $X_i$, where $[n] = \{1, \cdots, n\}$.*
*2. $\forall i \in [n], j \in N_i, \exists N_i \subset N_{ij} \subset [n]$, such that $W - X_{N_{ij}}$ is independent of $\{X_i, X_j\}$.*
*3. $\forall i \in [n], j \in N_i, k \in N_{ij}, \exists N_{ij} \subset N_{ijk} \subset [n]$ such that $W - X_{N_{ijk}}$ is independent of $\{X_i, X_j, X_k\}$.*

*Suppose further that for each $i \in [n], j \in N_i, k \in N_{ij}$,*

$$|X_i| \leq \beta, |N_i| \leq n_1, |N_{ij}| \leq n_2, |N_{ijk}| \leq n_3$$

*where $|\cdot|$ is the Euclidean norm of a vector. Then there exists a universal constant $C$ such that*

$$\sup_{\mathbb{O} \in \mathfrak{D}} \left| \mathbb{P}(W \in \mathbb{O}) - \mathbb{P}\left(\Sigma^{1/2} Z \in \mathbb{O}\right) \right| \leq Cp^{1/4}n \left\| \Sigma^{-1/2} \right\|^3 \beta^3 n_1 \left(n_2 + \frac{n_3}{\mathrm{d}}\right)$$

*where $Z$ is a $p$-dimensional Gaussian random vector preserving the covariance structure of $W$ and where $\mathfrak{D}$ denotes the collection of all the convex sets in $\mathbb{R}^p$.*

**Lemma G.4** (Lemma E.2 in Shi, Zhang, Lu & Song (2021)). *Suppose the conditions in Theorem 3.1 hold. We have as $N \to \infty$ that $\left\| \Sigma^{-1} \right\|_F \leq 3\bar{c}^{-1}, \|\Sigma\|_F = O(1), \|\Sigma - \Sigma\|_F = O_p\left\{ K^{1/2}(nT)^{-1/2} \log N \right\}, \left\| \widehat{\Sigma}^{-1} - \Sigma^{-1} \right\|_F = O_p\left\{ K^{1/2} N^{-1/2} \log N \right\}$ and $\left\| \widehat{\Sigma}^{-1} \right\|_F \leq 6\bar{c}^{-1}$ with probability approaching 1.*

**Lemma G.5** (Lemma E.3 in Shi, Zhang, Lu & Song (2021)). *Suppose the conditions in Theorem 3.1 hold. As $N \to \infty$, we have $\lambda_{\max}\left( T^{-1} \sum_{t=0}^{T-1} \mathbb{E}\xi_{0,t}\xi_{0,t}^\top \right) = O(1), \lambda_{\max}\left\{ N^{-1} \sum_{i=1}^{n} \sum_{t=0}^{T-1} \xi_{i,t}\xi_{i,t}^\top \right\} = O_p(1), \lambda_{min}\left( T^{-1} \sum_{t=0}^{T-1} \mathbb{E}\xi_{0,t}\xi_{0,t}^\top \right) \geq \bar{c}/2$ and $\lambda_{min}\left\{ N^{-1} \sum_{i=1}^{n} \sum_{t=0}^{T-1} \xi_{i,t}\xi_{i,t}^\top \right\} \geq \bar{c}/3$ with probability approaching 1.*

*Proof of Theorem 3.1.* By definition and the arguments in Section F.1, we have

$$
\hat{\theta} = \widehat{\boldsymbol{\beta}} - \boldsymbol{\beta}^* = \widehat{\Sigma}^{-1} \left[ \frac{1}{N} \sum_{i=1}^{n} \sum_{t=0}^{T-1} \xi_{i,t} \left\{ R_{i,t} - (\xi_{i,t} - \gamma \boldsymbol{U}_{i,t+1})^\top \boldsymbol{\beta}^* \right\} \right],
$$

$$
= \widehat{\Sigma}^{-1} \left[ \frac{1}{N} \sum_{i=1}^{n} \sum_{t=0}^{T-1} \xi_{i,t} \left\{ R_{i,t} - \Phi^\top (S_{i,t}) \beta_{A_{i,t}}^* + \gamma \sum_{a \in \mathcal{A}} \Phi^\top (S_{i,t+1}) \beta_a^* \pi (a \mid S_{i,t+1}) \right\} \right],
$$

$$
= \Sigma^{-1} \left( \frac{1}{N} \sum_{i=1}^{n} \sum_{t=0}^{T-1} \xi_{i,t}\varepsilon_{i,t} \right) + (\widehat{\Sigma}^{-1} - \Sigma^{-1}) \left( \frac{1}{N} \sum_{i=1}^{n} \sum_{t=0}^{T-1} \xi_{i,t}\varepsilon_{i,t} \right) + O(K^{-p/d}), \tag{S.9}
$$

where

$$
\varepsilon_{i,t} = R_{i,t} + \gamma \sum_{a \in \mathcal{A}} Q^* (\pi; S_{i,t+1}, a) \pi (a \mid S_{i,t+1}) - Q^* (\pi; S_{i,t}, A_{i,t}). \tag{S.10}
$$

Denote $\mathcal{G}_{i,t}$ as the sub-dataset $\{S_{i,t}, A_{i,t}\} \cup \{(R_{i,j}, A_{i,j}, S_{i,j})\}_{1 \leq j < t}$. By the Bellman equation and conditions (2.1) and (2.2), we have

$$
\mathbb{E} \left( \varepsilon_{i,t} \mid \mathcal{F}_{i,t} \right) = \mathbb{E} \left( \varepsilon_{i,t} \mid S_{i,t}, A_{i,t} \right) = 0
$$

Recall the definition $\xi_{i,t} = \xi(S_{i,t}, A_{i,t})$ in (3.5), we have for any $0 \leq t_1 < t_2 \leq T - 1$

$$
\mathbb{E}\varepsilon_{i,t_1}\varepsilon_{i,t_2}\xi_{i,t_1}^\top \xi_{i,t_2} = \mathbb{E} \left\{ \varepsilon_{i,t_1}\xi_{i,t_1}^\top \xi_{i,t_2} \mathbb{E} \left( \varepsilon_{i,t_2} \mid \mathcal{F}_{i,t_2} \right) \right\} = 0
$$

Therefore, for any $0 \leq t_1 < t_2 \leq T - 1$ and $1 \leq i_1 < i_2 \leq n$ we have $\mathbb{E}\varepsilon_{i_1,t_1}\varepsilon_{i_2,t_2}\xi_{i_1,t_1}^\top \xi_{i_2,t_2} = 0$ and

$$
\left\| \sum_{i=1}^{n} \sum_{t=0}^{T-1} \xi_{i,t}\varepsilon_{i,t} \right\|^2 = \sum_{i=1}^{n} \sum_{t=0}^{T-1} \mathbb{E}\varepsilon_{i,t}^2 \xi_{i,t}^\top \xi_{i,t} = n \sum_{t=0}^{T-1} \mathbb{E}\varepsilon_{0,t}^2 \xi_{0,t}^\top \xi_{0,t}
$$

By Assumption (A3) and Lemma F.2, we have

$$
\left\| \sum_{i=1}^{n} \sum_{t=0}^{T-1} \xi_{i,t}\varepsilon_{i,t} \right\|^2 \lesssim n \sum_{t=0}^{T-1} \mathbb{E}\xi_{0,t}^\top \xi_{0,t} \lesssim nT \sup_{s \in \mathcal{S}} |\Phi(s)|^2 = O(NK)
$$

By Markov's inequality, $N^{-1} \sum_{i=1}^{n} \sum_{t=0}^{T-1} \xi_{i,t}\varepsilon_{i,t} = O_p(\sqrt{K/N})$, and together with Lemma G.4, we have

$$
(\widehat{\Sigma}^{-1} - \Sigma^{-1}) \left( \frac{1}{N} \sum_{i=1}^{n} \sum_{t=0}^{T-1} \xi_{i,t}\varepsilon_{i,t} \right) = O_p \left( KN^{-1} \log N \right). \tag{S.11}
$$

In the following, we show the convex Gaussian approximation on $\Sigma^{-1} \left( \frac{1}{\sqrt{N}} \sum_{i=1}^{n} \sum_{t=0}^{T-1} \xi_{i,t}\varepsilon_{i,t} \right)$. Denote $\mathbf{Z}_N = \frac{1}{\sqrt{N}} \sum_{i=1}^{n} \sum_{t=0}^{T-1} \mathbf{z}_{i,t}$ and $\mathbf{z}_{i,t} = \xi_{i,t}\varepsilon_{i,t}$, define the truncated $\mathbf{z}_{i,t}$ as

$$
\bar{\mathbf{z}}_{i,t} = \begin{cases} \mathbf{z}_{i,t}, & |\mathbf{z}_{i,t}| \leq \pi_N \\ 0, & \text{otherwise.} \end{cases} \tag{S.12}
$$

Denote $\bar{\mathbf{Z}}_N = \sum_{i=1}^{n} \sum_{t=0}^{T-1} \bar{\mathbf{z}}_{i,t}/\sqrt{N}$ and $\bar{\mathbf{Z}}_N^* =: \bar{\mathbf{Z}}_N - \mathbb{E}\bar{\mathbf{Z}}_N$. Suppose $T/m = k \in \mathbb{N}$ w.l.o.g., and define

$$
\bar{\mathbf{z}}_{i,t}^{(m)} =: \mathbb{E}(\bar{\mathbf{z}}_{i,t} \mid \mathcal{F}_m(t)), \tag{S.13}
$$

where $\mathcal{F}_m(t) = \sigma(\zeta_{t-m+1}, \ldots, \zeta_t)$. Then $\bar{\mathbf{z}}_{i,k}^{(m)}, \bar{\mathbf{z}}_{i,j}^{(m)}$ are independent as long as $|k-j| > m$. Further let $\bar{\mathbf{Z}}_N^{(m)} =: \sum_{i=1}^n \sum_{t=0}^{T-1} \bar{\mathbf{z}}_{i,t}^{(m)}/\sqrt{N}$ and $\tilde{\mathbf{Z}}_N^{(m)} =: \bar{\mathbf{Z}}_N^{(m)} - \mathbb{E}\bar{\mathbf{Z}}_N^{(m)}$, then $\bar{\mathbf{Z}}_N^* - \tilde{\mathbf{Z}}_N^{(m)} = \bar{\mathbf{Z}}_N - \bar{\mathbf{Z}}_N^{(m)}$. Denote the covariance matrices $\Omega$ and $\Omega^{(m)}$ of $\mathbf{Z}_N$ and $\bar{\mathbf{Z}}_N^{(m)}$ respectively. Introduce a $p$-dimensional standard Gaussian random vector $\mathbf{G}$ and denote

$$\mathbf{G}_N = \Omega^{1/2}\mathbf{G}, \quad \tilde{\mathbf{G}}_N^{(m)} = \left(\Omega^{(m)}\right)^{1/2}\mathbf{G}$$

so that $\mathbf{G}_N$ and $\tilde{\mathbf{G}}_N^{(m)}$ preserve the covariance structure $\Omega, \Omega^{(m)}$, respectively. We then introduce the convex Kolmogorov distance to measure the convex distribution probability difference between $p$-dimensional random vectors $\mathbf{X}$ and $\mathbf{Y}$,

$$\mathcal{K}(\mathbf{X}, \mathbf{Y}) =: \sup_{\mathbb{O} \in \mathfrak{O}} |\mathrm{P}(\mathbf{X} \in \mathbb{O}) - \mathrm{P}(\mathbf{Y} \in \mathbb{O})|, \tag{S.14}$$

where $\mathfrak{O}$ is the collection of all the convex sets in $\mathbb{R}^p$. Combining (S.9) and (S.11), it suffices to show $\mathcal{K}(\mathbf{Z}_N, \mathbf{G}_N) = o(1)$. By Lemma G.2 and $|\nabla h_{A,\epsilon}| \le 2\epsilon^{-1}$, we can decompose the $\mathcal{K}(\mathbf{Z}_N, \mathbf{G}_N)$ as

$$\mathcal{K}(\mathbf{Z}_N, \mathbf{G}_N) \le 4K^{1/4}\epsilon_1 + \sup_{\mathbb{O} \in \mathfrak{O}} |\mathbb{E}[h_{\mathbb{O},\epsilon_1}(\mathbf{Z}_N) - h_{\mathbb{O},\epsilon_1}(\mathbf{G}_N)]|$$

$$\le K^{\frac{1}{4}}\epsilon_1 + \sup_{\mathbb{O} \in \mathfrak{O}}\left|\mathbb{E}\left[h_{\mathbb{O},\epsilon_1}(\mathbf{Z}_N) - h_{\mathbb{O},\epsilon_1}(\bar{\mathbf{Z}}_N^*)\right]\right| + \sup_{\mathbb{O} \in \mathfrak{O}}\left|\mathbb{E}\left[h_{\mathbb{O},\epsilon_1}(\bar{\mathbf{Z}}_N^*) - h_{\mathbb{O},\epsilon_1}(\tilde{\mathbf{Z}}_N^{(m)})\right]\right|$$

$$+ \sup_{\mathbb{O} \in \mathfrak{O}}\left|\mathbb{E}\left[h_{\mathbb{O},\epsilon_1}(\tilde{\mathbf{G}}_N^{(m)}) - h_{\mathbb{O},\epsilon_1}(\mathbf{G}_N)\right]\right| + \sup_{\mathbb{O} \in \mathfrak{O}}\left|\mathbb{E}\left[h_{\mathbb{O},\epsilon_1}(\tilde{\mathbf{Z}}_N^{(m)}) - h_{\mathbb{O},\epsilon_1}(\tilde{\mathbf{G}}_N^{(m)})\right]\right|,$$

$$\lesssim K^{\frac{1}{4}}\epsilon_1 + \frac{1}{\epsilon_1}\mathbb{E}\left|\mathbf{Z}_N - \bar{\mathbf{Z}}_N^*\right| + \frac{1}{\epsilon_1}\mathbb{E}\left|\bar{\mathbf{Z}}_N^* - \tilde{\mathbf{Z}}_N^{(m)}\right| + \frac{1}{\epsilon_1}\mathbb{E}\left|\tilde{\mathbf{G}}_N^{(m)} - \mathbf{G}_N\right|$$

$$+ \sup_{\mathbb{O} \in \mathfrak{O}}\left|\mathbb{E}\left[h_{\mathbb{O},\epsilon_1}(\tilde{\mathbf{Z}}_N^{(m)}) - h_{\mathbb{O},\epsilon_1}(\tilde{\mathbf{G}}_N^{(m)})\right]\right|. \tag{S.15}$$

Based on decomposition (S.15), for $q > 4$ and appropriate $m \asymp \log n$, we shall prove the following assertions as follows:

**(1)** Truncation error

$$\mathbb{E}\left|\mathbf{Z}_N - \bar{\mathbf{Z}}_N^*\right| = O(\sqrt{N}\pi_N^{1-q}\xi_{K,n}^q). \tag{S.16}$$

**(2)** M-decomposition error

$$\mathbb{E}\left|\bar{\mathbf{Z}}_n^* - \tilde{\mathbf{Z}}_n^{(m)}\right| = o(\sqrt{n}\pi_n^{1-q}\xi_{K,n}^q). \tag{S.17}$$

**(3)** Gaussian comparison

$$\mathbb{E}\left|\tilde{\mathbf{G}}_N^{(m)} - \mathbf{G}_N\right| = O\left(N\pi_N^{2-2q}\xi_{K,n}^{2q}\right). \tag{S.18}$$

**(4)** Gaussian approximation

$$\sup_{\mathbb{O} \in \mathfrak{O}}\left|\mathbb{E}\left[h_{\mathbb{O},\epsilon_1}(\tilde{\mathbf{Z}}_N') - h_{\mathbb{O},\epsilon_1}(\tilde{\mathbf{G}}_N^{(m)})\right]\right| = O\left(K^{1/4}N^{-1/2}\pi_N^3\log^2 N + \frac{1}{\epsilon_1}K^{1/4}N^{-1}\pi_N^6\log^4 N\right). \tag{S.19}$$

**Truncation error** In view of $\mathbb{E}\mathbf{Z}_N = 0$ and $\mathbf{z}_{i,t} - \bar{\mathbf{z}}_{i,t} = \mathbf{z}_{i,t}\mathbf{1}_{|\mathbf{z}_{i,t}|>\pi_N}$, we have for $q > 1$,

$$\mathbb{E}|\mathbf{Z}_N - \bar{\mathbf{Z}}_N^*| \le \mathbb{E}|\mathbf{Z}_N - \bar{\mathbf{Z}}_N| + |\mathbb{E}\mathbf{Z}_N - \mathbb{E}\bar{\mathbf{Z}}_N|$$

$$\le \frac{2}{\sqrt{N}}\mathbb{E}\left|\sum_{i=1}^n\sum_{t=0}^{T-1}\mathbf{z}_{i,t}\mathbf{1}_{|\mathbf{z}_{i,t}|>\pi_N}\right|$$

$$\le 2\sqrt{N}\mathbb{E}\left[|\mathbf{z}_{i,t}|\left(\frac{|\mathbf{z}_{i,t}|}{\pi_N}\right)^{q-1}\right]$$

$$= 2\sqrt{N}\pi_N^{1-q}\|\mathbf{z}_{i,t}\|_q^q, \tag{S.20}$$

which yields (S.16) using the fact $\|\mathbf{z}_{i,t}\|_q = O(\sup_s |\Phi(s)|) = O(\xi_{K,N})$ for any given $q \in \mathbb{N}^+$. Moreover, we can also have for $q > 1$,

$$\|\mathbf{Z}_N - \bar{\mathbf{Z}}_N^*\| \leq 2\sqrt{N} \left\{ \mathbb{E}\left[ |\mathbf{z}_{i,t}|^2 \left( \frac{|\mathbf{z}_{i,t}|}{\pi_N} \right)^{2q-2} \right] \right\}^{1/2}$$

$$= O(T_{n,q}), \tag{S.21}$$

where $T_{N,q} =: \sqrt{n}\pi_N^{1-q}\|\mathbf{z}_{i,t}\|_{2q}^q$.

**M-decomposition error**   Denote operator $\mathcal{P}^{(k)}\bar{\mathbf{z}}_{i,t} =: \mathbb{E}(\bar{\mathbf{z}}_{i,t}|\mathcal{F}_k(t)) - \mathbb{E}(\bar{\mathbf{z}}_{i,t}|\mathcal{F}_{k-1}(t))$ using the fact $\bar{\mathbf{z}}_{i,t} = \lim_{j \to \infty} \mathbb{E}(\bar{\mathbf{z}}_{i,t}|\mathcal{F}_{t+j}(t))$, we have on a richer space,

$$\bar{\mathbf{Z}}_N - \bar{\mathbf{Z}}_N^{(m)} = \frac{1}{\sqrt{N}} \sum_{i=1}^{n} \sum_{t=0}^{T-1} \sum_{j=m-t+1}^{\infty} \mathcal{P}^{(t+j)}\bar{\mathbf{z}}_{i,t} = \frac{1}{\sqrt{N}} \sum_{i=1}^{n} \sum_{j=m-T+1}^{\infty} \mathbf{R}_{i,T,j}, \tag{S.22}$$

where $\mathbf{R}_{i,T,j} =: \sum_{i=(m-j+1)\vee 1}^{T} \mathcal{P}^{(t+j)}\bar{\mathbf{z}}_{i,t}$. By Jensen's inequality, for $q > 1$,

$$\left\| \mathcal{P}^{(k)}\bar{\mathbf{z}}_{i,t} \right\|_q \leq \left\| \mathcal{P}^{(k)}\mathbf{z}_{i,t} \right\|_q = \{\mathbb{E}|\mathbb{E}(\mathbf{z}_{i,t}|\zeta_{t-k+1}, \ldots, \zeta_t) - \mathbb{E}(\mathbf{z}_{i,t}|\zeta_{t-k+2}, \ldots, \zeta_t)|^q\}^{1/q}$$

$$= \left\{ \mathbb{E}\left|\mathbb{E}\left[ \mathbb{E}(\mathbf{z}_{i,t}|\mathcal{H}_{t,k-1}) - \mathbb{E}(\mathbf{z}_i|\mathcal{H}_t) \Big| \mathcal{H}_{t-k+1} \right]\right|^q \right\}^{1/q} \lesssim \delta_\Phi(k-1, q). \tag{S.23}$$

Note that for given $i$, process $\{\mathbf{R}_{i,T,j}, j \geq m - n + 1\}$ is martingale difference with respect to filtration $\sigma(\zeta_{-j+1}, \zeta_{-j+2}, \ldots)$. If $q \geq 2$, by Burkholder's inequality, there exists constant $C_q > 0$ such that

$$\left\| \sum_{j=m-n+1}^{\infty} \mathbf{R}_{i,T,j} \right\|_q^2 \leq C_q \sum_{j=m-n+1}^{\infty} \|\mathbf{R}_{i,T,j}\|_q^2$$

$$\leq C_q \sum_{j=m-n+1}^{\infty} \left( \sum_{l=(m-j+1)\vee 1}^{n} \left\| \mathcal{P}^{(l+j)}\bar{\mathbf{z}}_{i,l} \right\|_q \right)^2. \tag{S.24}$$

Using the fact $\delta_\Phi(k,q) = O(\Delta_{K,N}^\alpha \chi^{\alpha k}\xi_{K,N})$ for any given $\alpha \in (0, 1)$, (S.23) yields

$$\left\| \mathcal{P}^{(k)}\bar{\mathbf{z}}_i \right\| = O(\xi_{K,N}\Delta_{K,N}^\alpha \chi^{\alpha k}). \tag{S.25}$$

Combining (S.22), (S.23), (S.24), and (S.25) elementary calculation yields

$$\|\bar{\mathbf{Z}}_N^* - \tilde{\mathbf{Z}}_N^{(m)}\| = \|\bar{\mathbf{Z}}_N - \bar{\mathbf{Z}}_N^{(m)}\| = \frac{1}{\sqrt{N}} \sum_{i=1}^{n} \left\| \sum_{j=m-n+1}^{\infty} \mathbf{R}_{i,T,j} \right\| = O(\xi_{K,N}\Delta_{K,N}^\alpha \chi^{\alpha m}). \tag{S.26}$$

Setting appropriate m-decomposition $m \asymp \log N$ (e.g., $m = \frac{(q-4)|\omega_1+\omega_0/12-1/6|\log N}{\alpha \log(1/\chi)}$ ), we have for $q > 4$,

$$\xi_{K,n}\Delta_{K,N}^\alpha \chi^{\alpha m} \ll T_{N,q} \tag{S.27}$$

with $\alpha < \min\{1, \frac{q-4}{2\omega_1'}|\omega_1 + \omega_0/12 - 1/6|\}$.

**Gaussian comparison**   For a matrix $\mathbf{A}$, denote $\|\mathbf{A}\|_F$ as the Frobenius norm of $\mathbf{A}$ i.e. $\|\mathbf{A}\|_F = \left(\text{tr}(\mathbf{A}^\top\mathbf{A})\right)^{1/2}$. Recall $\Omega = \mathbb{E}\left( \sum_{i=1}^{n} \sum_{t=0}^{T-1} \mathbf{z}_{i,t} \right) \left( \sum_{i=1}^{n} \sum_{t=0}^{T-1} \mathbf{z}_{i,t}^\top \right)/N$ and

$$\Omega^{(m)} =: \frac{1}{N}\mathbb{E}\left\{ \left[ \sum_{i=1}^{n} \sum_{t=0}^{T-1} \left( \bar{\mathbf{z}}_{i,t}' - \mathbb{E}\bar{\mathbf{z}}_{i,t} \right) \right] \left[ \sum_{i=1}^{n} \sum_{t=0}^{T-1} \left( \bar{\mathbf{z}}_{i,t}' - \mathbb{E}\bar{\mathbf{z}}_{i,t} \right)^\top \right] \right\}.$$

Consider the difference of covariance matrix between $\bar{\mathbf{Z}}_N^{(m)}$ and $\mathbf{Z}_N$ based on Frobenius norm, using the fact $\mathbb{E}\mathbf{Z}_N = 0$ and $\mathbb{E}\bar{\mathbf{Z}}_N^{(m)} = \mathbb{E}\bar{\mathbf{Z}}_N$,

$$\left\| \Omega - \Omega^{(m)} \right\|_F \leq \left\| \mathbb{E}\left(\mathbf{Z}_N - \bar{\mathbf{Z}}_N^{(m)}\right)(\mathbf{Z}_N)^\top \right\|_F + \left\| \mathbb{E}\bar{\mathbf{Z}}_N^{(m)}\left(\mathbf{Z}_N - \bar{\mathbf{Z}}_N^{(m)}\right)^\top \right\|_F$$
$$+ \left\| \mathbb{E}\left(\mathbf{Z}_N - \bar{\mathbf{Z}}_N\right)\mathbb{E}\left(\mathbf{Z}_N - \bar{\mathbf{Z}}_N\right)^\top \right\|_F. \tag{S.28}$$

By (S.21) and (S.26),

$$\left\| \mathbb{E}\left(\mathbf{Z}_N - \bar{\mathbf{Z}}_N^{(m)}\right)(\mathbf{Z}_N)^\top \right\|_F \leq \|\mathbf{Z}_N - \bar{\mathbf{Z}}_N^{(m)}\| \cdot \|\mathbf{Z}_N\| = O\left(T_{n,q}\xi_{K,n}\Delta_{K,n}^\alpha \chi^{\alpha m}\right), \tag{S.29}$$

and similarly, we also have

$$\left\| \mathbb{E}\bar{\mathbf{Z}}_N^{(m)}\left(\mathbf{Z}_N - \bar{\mathbf{Z}}_N^{(m)}\right)^\top \right\|_F \leq \|\mathbf{Z}_N - \bar{\mathbf{Z}}_N^{(m)}\| \cdot \|\bar{\mathbf{Z}}_N\| = O\left(T_{n,q}\xi_{K,n}\Delta_{K,n}^\alpha \chi^{\alpha m}\right). \tag{S.30}$$

Besides, by (S.20),

$$\left\| \mathbb{E}\left(\mathbf{Z}_N - \bar{\mathbf{Z}}_N\right)\mathbb{E}\left(\mathbf{Z}_N - \bar{\mathbf{Z}}_N\right)^\top \right\|_F \leq \left|\mathbb{E}(\mathbf{Z}_N - \bar{\mathbf{Z}}_N)\right|^2 = O\left(T_{n,q}^2\right). \tag{S.31}$$

Combining (S.29), (S.30), (S.31),

$$\left\| \Omega - \Omega^{(m)} \right\|_F = O\left(T_{n,q}^2 + T_{n,q}\xi_{K,n}\Delta_{K,n}^\alpha \chi^{\alpha m}\right). \tag{S.32}$$

By Assumption (A3) and Lemma G.5, we have $\inf_{s,a}\omega(s,a) \geq c_0^{-1}$ and

$$\lambda_{min}(\Omega) \geq c_0^{-1}\lambda_{min}(\frac{1}{T}\sum_{t=0}^{T-1}\mathbb{E}\xi_{0,t}\xi_{0,t}^\top) \gtrsim c_0^{-1}\bar{c}, \tag{S.33}$$

which yields

$$\lambda_{min}(\Omega^{(m)}) \geq \lambda_{min}(\Omega) + \lambda_{min}(\Omega - \Omega^{(m)}) \gtrsim c_0^{-1}\bar{c} - \left\| \Omega - \Omega^{(m)} \right\|_F > 0. \tag{S.34}$$

Combing (S.32), (S.34) and sub-multiplicativity of Frobenius norm, we have

$$\mathbb{E}\left|\tilde{\mathbf{G}}_N^{(m)} - \mathbf{G}_N\right| = \mathbb{E}\left|\left(\Omega^{1/2} - \left(\Omega^{(m)}\right)^{1/2}\right)\mathbf{G}\right|$$
$$\leq \left\| \Omega^{1/2} - \left(\Omega^{(m)}\right)^{1/2} \right\|_F = O\left(T_{n,q}^2 + T_{n,q}\xi_{K,n}\Delta_{K,n}^\alpha \chi^{\alpha m}\right). \tag{S.35}$$

By similar arguments in (S.27), $\xi_{K,n}\Delta_{K,n}^\alpha \chi^{\alpha m} = o(T_{n,q})$ by appropriate $m$ and $\alpha$, which yields (S.18).

**Gaussian approximation**  Plug $n_1 = m$, $n_2 = 2m$, $n_3 = 3m$, $\kappa = N^{-1/2}\pi_N$ into Lemma G.3 with (S.34), we have

$$\mathcal{K}\left(\tilde{\mathbf{Z}}_N^{(m)}, \tilde{\mathbf{G}}_N^{(m)}\right) = O\left(K^{1/4}N^{-1/2}\pi_N^3\log^2 n\right). \tag{S.36}$$

Moreover, in the proof of Lemma G.3, equation (4.23) in Fang (2016) yields

$$\sup_{\mathbb{O}\in\mathfrak{O}}\left|\mathbb{E}\left[h_{\mathbb{O},\epsilon_1}\left(\tilde{\mathbf{Z}}_N^{(m)}\right) - h_{\mathbb{O},\epsilon_1}\left(\tilde{\mathbf{G}}_N^{(m)}\right)\right]\right| \leq Cn\kappa^3 n_1 n_2 \epsilon_1^{-1}\left[K^{1/4}(\epsilon_1 + n_3\beta) + \mathcal{K}\left(\tilde{\mathbf{Z}}_N^{(m)}, \tilde{\mathbf{G}}_N^{(m)}\right)\right],$$

by (S.36),

$$\sup_{\mathbb{O}\in\mathfrak{O}}\left|\mathbb{E}\left[h_{\mathbb{O},\epsilon_1}\left(\tilde{\mathbf{Z}}_N^{(m)}\right) - h_{\mathbb{O},\epsilon_1}\left(\tilde{\mathbf{G}}_N^{(m)}\right)\right]\right| = O\left(K^{1/4}N^{-1/2}\pi_N^3\log^2 N + \frac{1}{\epsilon_1}K^{1/4}N^{-1}\pi_N^6\log^4 N\right). \tag{S.37}$$

Let $\pi_N = \sqrt{\kappa_1 \kappa_2}$, where

$$\kappa_1 =: N^{\frac{1}{2(q-1)}} K^{\frac{1}{4(q-1)}} \xi_{K,n}^{\frac{q}{q-1}}, \quad \kappa_2 =: N^{\frac{1}{6}} K^{-\frac{1}{12}} \log^{-\frac{2}{3}} N. \tag{S.38}$$

Combining (S.16), (S.17), (S.18), (S.19), and (S.15),

$$\mathcal{K}(\mathbf{Z}_N, \mathbf{G}_N) = O\left(K^{\frac{1}{4}}\epsilon_1 + \frac{1}{\epsilon_1}\left(N^{\frac{1}{2}}\pi_N^{1-q}\xi_{K,N}^q + N\pi_N^{2-2q}\xi_{K,n}^{2q} + K^{\frac{1}{4}}N^{-1}\pi_N^6 \log^4 n\right) + K^{\frac{1}{4}}N^{-\frac{1}{2}}\pi_N^3 \log^2 N\right).$$

Therefore, with appropriate $\epsilon_1$ and $q > 4$, we have when $K = o(n^{2/7-c})$ for any given $c > 0$,

$$\mathcal{K}(\mathbf{Z}_N, \mathbf{G}_N) = O\left(\sqrt{K^{\frac{1}{4}}N^{\frac{1}{2}}\pi_N^{1-q}\xi_{K,N}^q} + K^{\frac{1}{8}}N^{\frac{1}{2}}\pi_N^{1-q}\xi_{K,N}^q + K^{\frac{1}{4}}N^{-\frac{1}{2}}\pi_N^3 \log^2 N\right) = o(1). \tag{S.39}$$

Furthermore, combining Lemma G.4 and using the fact $K \ll \sqrt{N}/\log N$, by similar arguments in (E.29) of Shi, Wan, Chernozhukov & Song (2021), one can show $\|\hat{\Sigma}^{-1}\hat{\Omega}(\hat{\Sigma}^{\top})^{-1} - \Sigma^{-1}\Omega\Sigma^{-1}\|_F = o_p(1)$, which yields the validation of the Bootstrap algorithm from the Slutsky's theorem. □

### G.3 Proof of Proposition 3.2

*Proof.* It suffices to find $C_{\alpha,N}$ such that as $N \to \infty$,

$$\mathrm{P}\left(\sup_{s \in \mathcal{S}} |\mathbf{T}(s)^{\top}\mathbf{G}| \leq C_{\alpha,N}\right) \to 1 - \alpha, \tag{S.40}$$

where $\mathbf{T}(s) = \mathbf{l}(s)/|\mathbf{l}(s)| = U(s)^{\top}\Lambda^{1/2}/\sqrt{U(s)^{\top}\Lambda U(s)}$ and $\mathbf{G}$ is standard $p$-dimensional random vector. Denote manifold

$$\mathcal{M} =: \{\mathbf{T}(s) : s \in \mathcal{S}\}, \tag{S.41}$$

and let $\kappa_0$ be the volume of the manifold $\mathcal{M}$, and $\zeta_0$ be the area of the boundary $\partial\mathcal{M}$; Let $\kappa_2$ and $\zeta_1$ be measures of the curvature of $\mathcal{M}$ and $\partial\mathcal{M}$ respectively, and $m_0$ measures the rotation angles in the regions $\partial^2\mathcal{M}$.

By Proposition 3 in Sun & Loader (1994), for the $\alpha$ in (S.40), we have

$$\begin{aligned}
\alpha =&\kappa_0 \frac{\Gamma((d+1)/2)}{\pi^{(d+1)/2}}\mathrm{P}\left(\chi_{d+1}^2 > C_{\alpha,N}^2\right) + \frac{\zeta_0}{2}\frac{\Gamma(d/2)}{\pi^{d/2}}\mathrm{P}\left(\chi_d^2 > C_{\alpha,N}^2\right) \\
&+ \frac{\kappa_2 + \zeta_1 + m_0}{2\pi}\frac{\Gamma((d-1)/2)}{\pi^{(d-1)/2}}\mathrm{P}\left(\chi_{d-1}^2 > C_{\alpha,N}^2\right) \\
&+ O\left(C_{\alpha,N}^{d-4}\exp\left(-\frac{C_{\alpha,N}^2}{2}\right)\right),
\end{aligned} \tag{S.42}$$

where $\chi_d^2$ is the chi-square random variable with the degree of freedom $d$.

To bound the positive geometric quantities $\kappa_0, \zeta_0, \kappa_2, \zeta_1, m_0$ appearing in (S.42), we give the following formulations for numerical computation. For simplicity, we suppose $\mathcal{S} = [0,1]^d$ and the boundary $\partial\mathcal{S}$ consists of those points $s$ with exactly one component 0 or 1. The regions where two faces of $\partial\mathcal{S}$ meet are denoted $\partial^2\mathcal{S}$. Denote matrix $\mathbf{A} = (\mathbf{T}_1(s), \ldots, \mathbf{T}_d(s))$ where $\mathbf{T}_j(s) = \partial\mathbf{T}(s)/\partial x_j$ with $s = (x_1, \ldots, x_d)^{\top}$ and indicator vector $\mathbf{e}_j = (e_{j,1}, \ldots, e_{j,d})^{\top}$ such that $e_{j,j} = 1$ and $e_{j,k} = 0$ if $k \neq j$. By (3.2) and (3.3) in Sun & Loader (1994), the $\kappa_0$ and $\kappa_2$ can be computed as

$$\kappa_0 = \int_{\mathcal{S}} \det^{1/2}(\mathbf{A}^{\top}\mathbf{A})\mathrm{d}s, \tag{S.43}$$

$$\kappa_2 = \int_{\mathcal{S}}\left\{\frac{S(s)}{2} - \frac{d(d-1)}{2}\right\}\det^{1/2}(\mathbf{A}^{\top}\mathbf{A})\mathrm{d}s, \tag{S.44}$$

where

$$S(s) = 2\sum_{j=2}^{d}\sum_{k=1}^{j-1}[\alpha_{j,j}(s)^{\top}\alpha_{k,k}(s) - \alpha_{j,k}(s)^{\top}\alpha_{k,j}(s)],$$

$$\alpha_{k,j}(s)^\top = \mathbf{e}_k^\top \left(\mathbf{A}^\top\mathbf{A}\right)^{-1}\frac{\partial\mathbf{A}^\top}{\partial x_j}\left(I - \mathbf{A}\left(\mathbf{A}^\top\mathbf{A}\right)^{-1}\mathbf{A}^\top\right).$$

For $\zeta_1, \zeta_0$ measuring the boundary $\partial\mathcal{M}$, by (3.4) in Sun & Loader (1994) and the second and third equation on Page 1335 of Sun & Loader (1994),

$$\zeta_0 = \int_{\partial\mathcal{S}} \det{}^{1/2}(\mathbf{A}_*^\top\mathbf{A}_*)\mathrm{d}s, \tag{S.45}$$

$$\zeta_1 = \int_{\partial\mathcal{S}} \zeta_1(s)\det{}^{1/2}(\mathbf{A}_*^\top\mathbf{A}_*)\mathrm{d}s, \tag{S.46}$$

where indicator vector $\mathbf{e}_j^* = (e_{j,1}, \ldots, e_{j,d-1})^\top$ such that $e_{j,j} = 1$ and $e_{j,k} = 0$ if $k \neq j$.

$$\zeta_1(s) = -\sum_{j=1}^{d-1}(\mathbf{e}_j^*)^\top(\mathbf{A}_*^\top\mathbf{A}_*)^{-1}\frac{\partial\mathbf{A}_*^\top}{\partial s_j}\mathbf{U}_d(s),$$

$$\mathbf{U}_d(s) \asymp (I - \mathbf{A}_*(\mathbf{A}_*^\top\mathbf{A}_*)^{-1}\mathbf{A}_*^\top)\mathbf{T}_d(s),$$
$$\mathbf{A}_* = (\mathbf{T}_1(s), \ldots, \mathbf{T}_{d-1}(s)), \tag{S.47}$$

on the face $s \in \partial\mathcal{S}$ at which $s_d$ is maximized, with similar definitions for $\zeta_1(s), \mathbf{U}_j(s), \mathbf{A}_*$ on other faces where $s_j$ is maximized. Moreover, by the fifth and seventh equations on Page 1335 of Sun & Loader (1994),

$$m_0 = \int_{\partial^2\mathcal{S}} m_0(s)\det{}^{1/2}(\mathbf{A}_{**}^\top\mathbf{A}_{**})\mathrm{d}s, \tag{S.48}$$

where

$$m_0(s) = \cos^{-1}\left(\mathbf{U}_{d-1}(s)^\top\mathbf{U}_d(s)\right),$$
$$\mathbf{A}_{**} = (\mathbf{T}_1(s), \ldots, \mathbf{T}_{d-2}(s)), \tag{S.49}$$

at a point $s$ at the meeting of the faces $s_{d-1} = 1$ and $s_d = 1$, with similar definitions for $m_0(s)$ on other meetings of the two faces of $\partial\mathcal{S}$.

Denote $\widetilde{\Phi}(s) = \Lambda^{1/2}U(s)$ and $\partial_{s_j}\widetilde{\Phi}(s) = \partial\widetilde{\Phi}(s)/\partial s_j$, then basic calculation yields

$$0 \le |\mathbf{T}_j(s)|^2 = \frac{|\partial_{s_j}\widetilde{\Phi}(s)|^2}{|\widetilde{\Phi}(s)|^2} - \frac{\left|\left(\partial_{s_j}\widetilde{\Phi}(s)\right)^\top\widetilde{\Phi}(s)\right|^2}{|\widetilde{\Phi}(s)|^4} \le \frac{|\partial_{x_j}\widetilde{\Phi}(s)|^2}{|\widetilde{\Phi}(s)|^2}.$$

Using the fact $\det^{1/d}(\mathbf{A}^\top\mathbf{A}) \le \mathrm{tr}(\mathbf{A}^\top\mathbf{A})/d$ and $\mathrm{tr}(\mathbf{A}^\top\mathbf{A}) = \sum_{j=1}^d |\mathbf{T}_j(s)|^2$, by (S.43), (S.45), (S.48). Note that $|\widetilde{\Phi}(s)| \ge \sqrt{\lambda_{min}(\Lambda)}|\Phi(s)| \gtrsim n^{c_0}$ by condition (3.15), Assumptions (A2) and (A3), there exists constant $c_1 > 0$ such that

$$\kappa_0 \le \int_\mathcal{S}\left(\frac{1}{d}\mathrm{tr}(\mathbf{A}^\top\mathbf{A})\right)^{d/2}\mathrm{d}s \lesssim \int_\mathcal{S}\left(\sum_{j=1}^d|\mathbf{T}_j(s)|^2\right)^{d/2}\mathrm{d}s \lesssim \int_\mathcal{S}\frac{|\nabla\widetilde{\Phi}(s)|^d}{|\widetilde{\Phi}(s)|^d}\mathrm{d}s = O(N^{c_1}),$$

and similarly, $\zeta_0 = O(N^{c_1})$, $m_0 = O(N^{c_1})$ since $\mathrm{tr}(\mathbf{A}_{**}^\top\mathbf{A}_{**}) \le \mathrm{tr}(\mathbf{A}_*^\top\mathbf{A}_*) \le \mathrm{tr}(\mathbf{A}^\top\mathbf{A})$.

For $\kappa_2$, note that matrix $\mathbf{A}(\mathbf{A}^\top\mathbf{A})^{-1}\mathbf{A}^\top$ is idempotent, thus

$$|\alpha_{k,j}(s)| \le \left|\frac{\partial\mathbf{A}}{\partial s_j}(\mathbf{A}^\top\mathbf{A})^{-1}\mathbf{e}_k\right| \le \frac{1}{\lambda_{min}(\mathbf{A}^\top\mathbf{A})}\left|\frac{\partial\mathbf{A}}{\partial s_j}\right|$$

By condition (3.15), for some constant $c_2 > 0$,

$$\kappa_2 \lesssim N^{c_1}\int_\mathcal{S}\frac{1}{\lambda_{min}(\mathbf{A}^\top\mathbf{A})}\left|\frac{\partial\mathbf{A}}{\partial s_j}\right|\mathrm{d}s + O(N^{c_1}) = O(N^{c_1+c_2}).$$

thus $\kappa_2 = O(N^{c_3})$ and similarly, we have $\zeta_1 = O(N^{c_3})$ for some constant $c_3 > 0$.

To sum up, there exists constant $\bar{c} > 0$ such that

$$\max\{\kappa_0, \zeta_0, \kappa_2, \zeta_1, m_0\} = O(N^{\bar{c}}). \tag{S.50}$$

By Theorem 6 in Zhang & Zhou (2020), for constants $\tilde{c}, \tilde{C}, \bar{C} > 0$, the tail bounds of $\chi_d^2$

$$\tilde{c} \exp\left(-\tilde{C}x\right) \leq \mathbb{P}(\chi_d^2 - d \geq x) \leq \exp\left(-\bar{C}x\right), \forall x > d, \tag{S.51}$$

Combining (S.50), (S.42), (S.51) and the fact $\alpha$ is a fixed value, we have $n^{\bar{c}} \exp(-\tilde{C}C_{\alpha,N}^2) \gtrsim 1$, which implies that $C_{\alpha,N} = O(\sqrt{\log N})$. On the other hand, by condition (3.15), there exists constant $\underline{c} > 0$, such that

$$\kappa_0 \geq \int_{\mathcal{S}} \lambda_{min}^{d/2}(\mathbf{A}^\top \mathbf{A}) \mathrm{d}s \gtrsim N^{\underline{c}}.$$

Combining (S.42), (S.51) and the fact $\alpha$ is a fixed value, we have $N^{\underline{c}} \exp(-\tilde{C}C_{\alpha,N}^2) \lesssim 1$, which shows $C_{\alpha,N} \gtrsim \sqrt{\log N}$. Combining (S.40),(3.11), and (3.4), we have appropriate $C_{\alpha,N} \asymp \log^{1/2} N$ such that (3.16) holds. □

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
