# OpenReview forum: "Simultaneous Statistical Inference for Off-Policy Evaluation in Reinforcement Learning"
_NeurIPS.cc/2025/Conference — NeurIPS 2025 poster_

### Official Review · Reviewer_SiUk · 2025-06-30

**Clarity:** 2
**Significance:** 2
**Originality:** 3
**Rating:** 4
**Confidence:** 4

**Summary:**

The paper proposes methods for simultaneous inference for off-policy evaluation in reinforcement learning. The primary problem addressed is the quantification of global uncertainty across an infinite or continuous initial state space. Existing methods primarily focus on point estimates or pointwise confidence intervals (CIs), which, when applied to multiple states inflate the overall significance level, especially in infinite state spaces. The proposed solution is the construction of asymptotically correct simultaneous confidence regions (SCRs) that cover the true policy value functions over the entire state space at a given significance level. The effectiveness of this approach is demonstrated through numerical simulations and analysis of the real-world OhioT1DM dataset

**Questions:**

1. Can the authors include a detailed literature on offline RL, highlighting the challenges in evaluation and learning? Especially, how to quantify the goodness of the dataset, when is learning possible etc.?
2. What is $L(s)$ in (3.1)?
3. How reasonable is it to assume access to an offline dataset in practice that permits learning and evaluation well enough for reliable confidence set computation?
4. Can the authors elaborate the derivation of (3.13)?
5. Can the authors explain the intuition behind Assumptions A1-A3.? How reasonable are the assumptions in the offline setting?
6. Can the authors provide a discussion of Theorem 3.1? The convergence in space-time to the gaussian process somehow gets absorbed into convergence in space? How are the asymptotic guarantees translating into finite time and space results?


Minor:
Assumption on Eq (2.2) is unclear.

**Ethical Concerns:**

["NO or VERY MINOR ethics concerns only"]

**Final Justification:**

The authors clarified all the concerns.

**Limitations:**

Yes

**Quality:**

3

**Strengths And Weaknesses:**

Strengths:
1. The SCRs developed are good in terms of efficiency, with widths exceeding those of pointwise CIs by only a logarithmic factor.
2. Empirical studies show that the choice and number of basis functions do not impact method's performance, enhancing its practical applicability.


Weaknesses:
1. Key assumptions are strong and unmotivated.
2. The offline setting is under explained, especially the challenges in offline RL.
3. It's unclear how to reconcile asymptotic results with the offline setting. The experimental demonstrations are unclear.

---

> ### Author Rebuttal · Authors · 2025-07-31
>
> We sincerely appreciate the time and effort you dedicated to reviewing our manuscript and providing insightful feedback. Below, we address each of your comments in detail.
>
> Q1&W2. **Regarding the offline RL, challenges, and the goodness of dataset.**
>
> We will include [1]-[5] as literature on offline RL and highlight the challenges in offline RL. The following argument will also apear in the revision. Unlike online environments (e.g. Go, MiniGrid), collecting data with online interaction in many real-world applications (e.g. healthcare, autonomous driving) can be expensive or even dangerous. This limitation prevents the widespread adoption of traditional online RL methods. As an alternative, offline RL has been proposed to effectively utilize pre-collected experience data without requiring any online interaction.
>
> However, this offline approach faces several challenges that make it different from online RL. For learning, the key challenges are:
> - Distributional shift: The data generated by the behavior policy (used to collect the dataset) might mismatch the target policy (which is learned), which leads to poor performance or overfitting.
> - Sample inefficiency: The learning must rely entirely on the static dataset, which means 'exploration' (as in online RL’s exploration-exploitation trade-off) is not possible during the learning process.
>
> For evaluation, offline RL faces the difficulty of **estimating policy performance without online deployment**, necessitating off-policy evaluation (OPE)—the focus of our work. One of the core challenges in OPE is the gap between estimated off-policy evaluation and real-world performance. To address this, recent attempts, including pointwise CIs (as we concluded in the related work section), have been proposed to quantify the gap between offline assessments and real-world deployment through probabilistic measures. Our work strengthens the statistical tools from pointwise to global scale, providing more reliable guarantees for decision-making.
>
> For the goodness of the dataset, the quality of an offline dataset depends critically on several factors:
> - State-space coverage: Breadth of visited states
> - Reward signal diversity: Range of observed returns
> - Action variety: Exploration of possible actions
> -  Algorithm compatibility: Match between data distribution and RL method
>
> As noted in [2], formalizing a non-trivial "sufficiency" condition for dataset quality remains an open problem. In our framework, we distill these complex considerations into Assumptions (A1)-(A3) (detailed in response Q5&W1). Under these assumptions, the goodness of the dataset can be roughly quantified by the sample size $N$. This simplification enables tractable theoretical analysis.
>
> Q2. **Regarding the $L(s)$ in (3.1).**
>
> $L(s)$ is a user-chosen scaling factor which shapes our SCR for different states $s\in\mathcal{S}$ ($L(s)>0$), especially the width. CIs in OPE consider $L(s)=\sqrt{U_\pi(s)^\top \Lambda_\pi U_\pi(s)}$ since the CLT theorem gaurantees $\sqrt{N}(\hat{V}(\pi;s)-V(\pi;s))/L(s)\to_d N(0,1)$ for fixed $s$.
>
> In our framework, the choice of $L(s)$ can be very flexible ($L(s)=\sqrt{U_\pi(s)^\top \Lambda_\pi U_\pi(s)}$ or simply $L(s)=1$). This is achieved by applying a stronger convex Gaussian approximation theorem (Theorem 3.1) instead of the simple CLT (the detailed explanation is put in response Q6&W3).
>
> Q3. **Regarding the accessibility of the offline dataset for reliable results.**
>
> Our theorems ensure that the SCRs are reliable when the sample size $N$ is sufficiently large. A large offline dataset is accessible in many real-world settings since
> - Offline RL leverages existing datasets (e.g., possibly historical logs, human demonstrations, or prior experiments).
> - Many domains (healthcare, education) have abundant offline data.
>
> Additionally, our theoretical reliability is built upon our Assumptions, which are mild for most cases and no stronger than those in serial theoretical work on offline RL (detailed in response Q5&W1). Thank you again for your comment, and the discussion on the access to offline data will be included in our revised paper.
>
> Q4. **Regarding the derivation of (3.13).**
>
> On the one hand, we write the probability in the first line of (3.13) by a more specific formulation:
> $$
> \mathrm{P}\left( \mathbf{Z}\_\pi\in \left\\{\theta\in \mathbb{R}^{mK}: \left|\Phi(s)^{\top} \sum_{a \in \mathcal{A}} \pi(a \mid s) \theta\right| \leq C_{\alpha, N} L(s), \forall s \in \mathcal{S}\right\\} \right)
> $$
> On the other hand, the second line in (3.13) can be formulated as
> $$
> \mathrm{P}\left( \mathbf{Z}\_\pi\in \left\\{ \theta\in \mathbb{R}^{mK}:  \sup_{s \in \mathcal{S}}\left|\frac{\Phi(s)^{\top} \sum_{a \in \mathcal{A}} \pi(a \mid s) \theta}{L(s)}\right| \leq C_{\alpha, N} \right\\} \right)
> $$
> Note that $L(s)>0$ and one can verify
> $$
> \left|\Phi(s)^{\top} \sum_{a \in \mathcal{A}} \pi(a \mid s) \theta\right| \leq C_{\alpha, N} L(s), \forall s \in \mathcal{S} \iff \sup_{s \in \mathcal{S}}\left|\frac{\Phi(s)^{\top} \sum_{a \in \mathcal{A}} \pi(a \mid s) \theta}{L(s)}\right| \leq C_{\alpha, N},
> $$
> thus, the derivation in (3.13) is correct.
>
> Q5&W1. **Regarding the intuition and rationality for Assumptions A1-A3.**
>
> Assumptions A1-A3 are very mild assumptions and serve as the minimal requirement for the goodness of the offline dataset to support feasible evaluation.
> - Assumption (A1) requires the density $\mu(s)$ for the invariant distribution to be uniformly bounded away from $0$ and $\infty$ so that the Markov chain would not be trapped in a small subset of the entire space. The density $\nu_0(s)$ for the initial state should also follow the boundary condition so that every state is possible to be the initial state.
> - Assumption (A2) relaxes the condition on sample size. Previous work (e.g., [5]) requires the number of trajectories $n\to\infty$. (A2) additionally allows fixed $n$, but length $T\to \infty$ when the action variety is sufficiently large on each chain. The geometrical ergodic for the Markov chain is technical for theoretical deduction and commonly assumed for deriving limit theory.
> - Assumption (A3) requires the reward signal diversity. The $\omega_\pi(s,a)\geq c_0^{-1}$ requires that the reward random variable is nondegenerate (not always the same). $P(\max_{0\leq t\leq T-1}|R_{0,t}|\leq c_0)=1$ means the rewards are bounded.
>
> These assumptions are the same as those needed to establish pointwise CIs in OPE as discussed in Remark 3.1.
>
> Q6&W3. **Regarding the discussion of Theorem 3.1.**
>
> Theorem 3.1 is our main theorem, which provides a higher convex Gaussian approximation towards $\hat{\theta}\_\pi=\hat{\beta}\_\pi-\hat{\beta}\_\pi\^*$ in the OPE estimation error $\hat{V}(\pi;s)-V(\pi;s)=\Phi(s)^\top \sum_{a\in\mathcal{A}}\pi(a|s)\hat{\theta}\_\pi$.
> This is a different approach from the previous pointwise CI results obtained by the simple CLT. The CLT focuses on deriving a limiting theorem on the **distribution function** of inner product value $\Phi(s)^\top \sum_{a\in\mathcal{A}}\pi(a|s)\hat{\theta}\_\pi$ for fixed $s\in\mathcal{S}$. The generalization from fixed $s\in\mathcal{S}$ to arbitrary $s\in\mathcal{S}$ is not straightforward, since one needs to control
> $$
>     \left|\mathrm{P}\left(\left|\frac{\Phi(s)^\top \sum_{a\in\mathcal{A}}\pi(a|s)\sqrt{N}\hat{\theta}\_\pi}{L(s)}\right|\leq t,\forall s\in\mathcal{S}\right) - \mathrm P\left(\left|\frac{\Phi(s)^\top \sum_{a\in\mathcal{A}}\pi(a|s)\mathbf{Z}\_\pi}{L(s)}\right|\leq t,\forall s\in\mathcal{S}\right)\right|
> $$
> where $\mathbf Z\_\pi$ is a random Gaussian vector.
> Note that for both scenarios of finite collection of $s$  and all $s\in \mathcal S$, the above can be rewritten as $\Delta\_{\mathbb O}=\left|\mathrm{P}\left(\sqrt{N} \hat{\theta}\_\pi \in \mathbb{O}\right)-\mathrm{P}\left(\mathbf{Z}\_\pi \in \mathbb{O}\right)\right|$ for some covex region $\mathbb O$ (see Remark 3.2 for details). When considering finite sample properties, we provide a bound concerning sample size $N$ and basis number $K$ from the proof of Theorem 3.1 (details in (S.37), (S.38), and (S.39) in the supplement),
> $$
> \sup_{\mathbb{O} \in \text{all convex sets} } \Delta\_{\mathbb{O}}
> \leq C\left(\sqrt{K^{\frac{1}{4}} N^{\frac{1}{2}} \pi_N^{1-q} \xi_{K, N}^q}+K^{\frac{1}{8}} N^{\frac{1}{2}} \pi_N^{1-q} \xi_{K, N}^q+K^{\frac{1}{4}} N^{-\frac{1}{2}} \pi_N^3 \log ^2 N\right).
> $$
> In the proof of Theorem 3.1, we also verify that the bound can converge to $0$ as $N\to \infty$ when $K$ is smaller than $N^{2/7}$. In this way we address the space-time issues in the Gaussian process in Theorem 3.1. Our convex Gaussian approximation focuses on the probability of the random vector $\hat{\theta}_\pi$ **on all convex sets** in $\mathbb{R}^{mK}$.  We will add a new remark in revision for the explanation.
>
> W3, part II. **Regarding the experimental demonstrations.**
>
> Due to the page limit, please kindly see our Q2bW2 response to Reviewer MvGP for our demonstration on OhioT1DM experiment.
>
> Minor. **Regarding Assumption on Eq (2.2).**
>
> Eq (2.2) assumes the conditional mean of the reward $R_{0,t}$ is independent of $(S\_{0,k},R\_{0,k},A\_{0,k})\_{k<t}$, i.e.
> $$
> E(R\_{0,t}|S\_{0,t}=s,A\_{0,t}=a,(S\_{0,k},R\_{0,k},A\_{0,k})\_{k<t})=E(R\_{0,t}|S\_{0,t}=s,A\_{0,t}=a)=r(s,a)
> $$
> for some reward function $r$.
>
> [1] Fujimoto S, Gu S S. A minimalist approach to offline reinforcement learning.
>
> [2] Levine S, Kumar A, Tucker G, et al. Offline reinforcement learning: Tutorial, review, and perspectives on open problems.
>
> [3] Prudencio R F, Maximo M R O A, Colombini E L. A survey on offline reinforcement learning: Taxonomy, review, and open problems.
>
> [4] Shi, C. (2025). Statistical inference in reinforcement learning: A selective survey.
>
> [5] Jiang, N. Li, L. (2016) Doubly robust off-policy value evaluation for reinforcement learning.

---

> > ### Comment · Reviewer_SiUk · 2025-08-01
> > **Post rebuttal**
> >
> > Thank you for your time and your response. The response did help clarify, but did not significantly change my evaluation owing to the concerns that still remain regards to the requirement of large offline datasets. So I choose to retain my score.

---

> ### Author Response · Authors · 2025-08-02
> **Response to SiUk for the large offline dataset concern**
>
> Thank you for acknowledging our clarifications. We respectfully emphasize that offline RL was specifically created to take advantage of large historical datasets. This isn't simply an added feature, but rather represents a core objective of the methodology. The method exists precisely for situations where we can't do online interaction (like in healthcare due to safety concerns) but do have existing data to learn from.
>
> Our empirical results demonstrate reliable performance of our large sample theory with dataset sizes around N=2000, and outperform modified classic methods (such as SAVE) when the sample size is larger. This scale is actually quite modest compared to typical offline RL applications. For example, the HiRID [1] and MIMIC-III [2] datasets both contain extensive ICU records collected over multiple years, with $N>10^8$. These datasets are widely used in medical AI research and are notable for their large volume and richness in temporal resolution. In the field of computer science, there are also many large-scale datasets available, see for example, D4RL (Datasets for Deep Data-Driven Reinforcement Learning), where most datasets have $N > 10^8$. Notice that the above are examples of publicly available data sets. Therefore, our approach can be reliably applied to evaluate policies across a wide range of scenarios.
>
> From a theoretical perspective, we understand your concern about sample size may stem from the fact that traditional asymptotic frameworks for evaluating maximum deviation lead to a Gumbel distribution (from extreme value theory), which exhibits slow convergence rates. Therefore, we stress that our theoretical results are built upon non-asymptotic results, which do not involve any convergence from extreme value theory in statistics.  In literature, even if the explicit critical value can be obtained from the limiting distribution derived from extreme value theory, one might still prefer bootstrap inference for better finite sample performance. Similarly, in our case, the bootstrap procedure for widths of the proposed SCR in our methods results in good performance under moderate sample size, as supported by simulations.
>
> We hope this evidence reassures you that our dataset requirements are not only realistic but also conservative compared to common practice.
>
> [1] HiRID: Faltys, M., Zimmermann, M., Lyu, X., Hüser, M., Hyland, S., Rätsch, G., & Merz, T. (2021). HiRID, a high time-resolution ICU dataset (version 1.1.1). PhysioNet. RRID:SCR_007345.
>
> [2] MIMIC-III: Johnson, A., Pollard, T., & Mark, R. (2016). MIMIC-III Clinical Database (version 1.4). PhysioNet. RRID:SCR_007345.

---

> > ### Comment · Reviewer_SiUk · 2025-08-07
> > **post-rebuttal**
> >
> > Thanks for the details regarding the size requirements. I will change my score to reflect this, with the understanding that the authors will include all these additional details that clarify and support as a part of the final submission.

---

> > > ### Author Response · Authors · 2025-08-08
> > >
> > > Thank you for your constructive feedback and for agreeing to revise your score. We will ensure that the requested points are added to the manuscript accordingly.

---

### Official Review · Reviewer_ZaNN · 2025-06-30

**Clarity:** 3
**Significance:** 2
**Originality:** 2
**Rating:** 4
**Confidence:** 3

**Summary:**

This paper tackles the fundamental problem of quantifying uncertainty in Off-Policy Evaluation (OPE) across entire state spaces, moving beyond the limitations of traditional pointwise confidence intervals. This is crucial for deploying RL policies with assured reliability.

**Questions:**

Please see the weakness.

**Ethical Concerns:**

["NO or VERY MINOR ethics concerns only"]

**Final Justification:**

I believe this paper has justified my concerns. I incline to give this paper accept.

**Limitations:**

Please see the weakness.

**Quality:**

2

**Strengths And Weaknesses:**

Strength:
1. The framework offers the first theoretically justified simultaneous inference for OPE, providing valid inference over the entire state space and effectively addressing the long-standing multiple testing problem in continuous or infinite state spaces. Its efficiency is notable, with SCR widths exceeding pointwise CIs by only a logarithmic factor $\sqrt{\log(N)}$, indicating near-optimal performance.
2. The multiplier bootstrap algorithm enhances practicality by obviating the need to estimate complex joint distributions, and the flexible scaling factor $L(s)$ allows for customization to specific application needs.

Weakness:
1. The constraint on $K$ relative to $N$($K = O(N^{2/7}(\log(N))^{-1})$) creates a tension. Achieving high approximation accuracy for complex Q-functions often demands a large $K$, which might conflict with the rate required for valid asymptotic inference, especially in high-dimensional settings.
2. The validity of the derived logarithmic rate for SCR width depends on specific geometric properties of the chosen basis functions (conditions 3.15). Verifying these properties for novel or highly complex basis function choices can be challenging in practice.
3. The "asymptotically correct" nature of the framework implies its validity for very large sample sizes. The paper's empirical results are promising, but exploration of its finite-sample behavior under various data characteristics (e.g., small $N$, high noise, non-Gaussian errors) is not presented in the paper.
4. The paper primarily compares its novel framework against Bonferroni-corrected pointwise confidence intervals. The absence of empirical comparisons against other advanced simultaneous inference techniques (if any exist or emerge in related fields) limits the understanding of its relative performance within the broader state-of-the-art for global inference.

---

> ### Author Rebuttal · Authors · 2025-07-31
>
> We sincerely appreciate your thorough review and insightful feedback on our manuscript. Thank you for recognizing the significance of our work in advancing uncertainty quantification for OPE in RL. We are particularly grateful for your acknowledgment of our framework's contributions regarding its efficiency, practical applicability, and flexibility. We have carefully considered all of your comments. In what follows, we address each of the questions you raised in detail.
>
> W1. **Regarding the constraint on $K$.**
>
> The condition $K=O(N^{2/7}\log^{-1}N)$ is imposed to ensure the convex Gaussian approximation error in Theorem 3.1 can attain $o(1)$.
> In contrast to the pointwise CLT ($K=O(N^{1/2}\log^{-1}N)$), this condition narrows the range of $K$ but does not lead to conflict in most cases.
>
> Take a simple case where the state space $\mathcal{S}$ is one-dimensional$(d=1)$ and the true Q-function lies in a H\"older space with smoothness $p=2$. On the one hand, the mathematical approximation theorem ensures approximation accuracy for the Q-function is of rate $\epsilon_K=O(K^{-2})$ (details included in Section D.1 of the appendix). On the other hand, our Proposition 3.1 shows our SCR width is of rate $O(N^{-1/2}\log^{1/2} N)$. Therefore, we wish the $K$ is sufficiently large so that $\epsilon_K$ is negligible compared to our SCR, i.e., $\epsilon_K\ll N^{-1/2}\log^{1/2} N$. This yields $K\gg N^{1/4}\log^{-1/4} N$ and does not conflict with $K=O(N^{2/7}\log^{-1}N)$ so that we can find a feasible $K=N^{\alpha}$ for some $\alpha\in (1/4,2/7)$ to support valid asymptotic inference.
>
> For the high-dimensional setting where $d\to \infty$, the curse of dimensionality is involved in the asymptotic rate. To the best of our knowledge, constructing confidence intervals/regions for the high-dimensional data in off-policy evaluation is still an open problem, which could be a future work.
>
>
> W2. **Regarding the geometric properties of the chosen basis functions.**
>
> We appreciate your insightful observation. Indeed, the validity of the logarithmic rate for SCR width relies on the geometric properties specified in Condition (3.15). While these properties can be rigorously verified for commonly used basis functions (e.g., Legendre polynomials, splines, and Fourier series in Section D of the supplement), extending the verification to arbitrary or highly complex bases may require additional substantial theoretical work.
>
> In practice, most applications employ such standard bases due to their well-characterized properties and computational tractability. Therefore, for the current analysis, we focus on establishing guarantees for those widely adopted bases that possess the geometric properties, and leave the generalizations to broader function classes as future work.
>
> W3. **Regarding the exploration of finite-sample behaviors under various data characteristics.**
>
> Thank you for your question.  Taking Scenario 2 (i.e., Table 1) as an illustrative example, we conducted additional simulations for both smaller and larger values of $N$. We kindly refer you to our **second** response to Reviewer MvGP for the results.  Moreover, for high noise and non-Gaussian errors, we modified the state transition rule, specifically the $z_{0,t}$ in line 268, to be an i.i.d. two-dimensional $t(8)$ random variable, while keeping the rest unchanged. The results are shown in Table R2. These results demonstrate that for noise with heavier tails, our coverage and length remain robust.
>
> **Table R2: Results for Scenario 2 with $t(8)$ noises across 500 repetitions.. Format: ECP(AL).**
>
> | $n$  | $T$  | Legendre SCR | Legendre SAVE (Bonferroni) | Legendre SAVE (Sidak) | Spline SCR | Spline SAVE (Bonferroni) | Spline SAVE (Sidak) |
> |------|------|--------------|------------------------------|--------------------------|------------|----------------------------|------------------------|
> | 30   | 50   | 0.940(4.706) | 0.934(4.431)                 | 0.980(4.424)             | 0.936(3.716) | 0.934(3.967)               | 0.986(3.961)           |
> | 50   | 30   | 0.944(2.721) | 0.932(3.316)                 | 0.990(3.311)             | 0.934(3.694) | 0.932(3.959)               | 0.988(3.953)           |
> | 40   | 50   | 0.946(3.349) | 0.934(3.658)                 | 0.992(3.652)             | 0.952(3.632) | 0.950(3.755)               | 0.990(3.749)           |
> | 50   | 40   | 0.932(2.876) | 0.928(3.279)                 | 0.990(3.274)             | 0.944(3.104) | 0.942(3.393)               | 0.988(3.388)           |
> | 50   | 50   | 0.934(3.488) | 0.926(3.448)                 | 0.986(3.443)             | 0.934(2.776) | 0.932(3.048)               | 0.994(3.044)           |
> | 50   | 150  | 0.948(2.333) | 0.944(2.498)                 | 0.988(2.494)             | 0.948(2.415) | 0.944(2.524)               | 0.988(2.520)           |
> | 50   | 200  | 0.942(2.012) | 0.946(2.168)                 | 0.984(2.165)             | 0.936(2.078) | 0.934(2.184)               | 0.988(2.181)           |
> | 50   | 250  | 0.932(1.962) | 0.934(2.111)                 | 0.994(2.108)             | 0.950(2.330) | 0.944(2.332)               | 0.986(2.329)           |
> | 200  | 70   | 0.946(1.970) | 0.996(2.024)                 | 0.996(2.021)             | 0.944(2.176) | 0.944(2.196)               | 0.980(2.192)           |
> | 250  | 70   | 0.948(1.644) | 0.980(1.784)                 | 0.980(1.780)             | 0.944(1.943) | 0.944(1.968)               | 0.982(1.965)           |
> | 300  | 70   | 0.930(1.495) | 0.986(1.625)                 | 0.986(1.623)             | 0.948(1.770) | 0.944(1.800)               | 0.980(1.797)           |
>
>
>
>
>
> W4. **Regarding comparisons against baseline other than Bonferroni correction.**
>
> Thank you for your question. In addition to the Bonferroni correction, we also compared our method with SAVE using Sidak correction [1], and the results are presented in Table R1 (we also paste it below for your convenience), and Table R2 in the previous question. The results show that, while ensuring coverage (ECP), our method achieves the shortest averaged length (AL).
>
> **Table R1: Results for Scenario 2 across 500 repetitions.. Format: ECP(AL).**
>
> | $n$  | $T$  | Legendre SCR | Legendre SAVE (Bonferroni) | Legendre SAVE (Sidak) | Spline SCR | Spline SAVE (Bonferroni) | Spline SAVE (Sidak) |
> |------|------|--------------|------------------------------|--------------------------|------------|----------------------------|------------------------|
> | 30   | 50   | 0.926(8.472) | 0.982(9.445)                 | 0.980(9.431)             | 0.936(9.452) | 0.978(9.793)               | 0.976(9.778)           |
> | 50   | 30   | 0.946(9.553) | 0.970(10.492)                | 0.968(10.476)            | 0.922(11.101) | 0.942(11.131)              | 0.942(11.115)          |
> | 40   | 50   | 0.924(7.225) | 0.976(8.193)                 | 0.976(8.181)             | 0.938(8.087) | 0.976(8.606)               | 0.974(8.593)           |
> | 50   | 40   | 0.944(7.138) | 0.990(8.136)                 | 0.990(8.124)             | 0.930(7.247) | 0.984(7.753)               | 0.984(7.741)           |
> | 50   | 50   | 0.942(7.299) | 0.978(8.249)                 | 0.978(8.237)             | 0.930(8.156) | 0.966(8.630)               | 0.966(8.617)           |
> | 50   | 150  | 0.952(6.985) | 0.966(7.402)                 | 0.974(7.393)             | 0.934(5.771) | 0.978(6.080)               | 0.972(6.075)           |
> | 50   | 200  | 0.944(5.978) | 0.978(6.436)                 | 0.974(6.435)             | 0.950(7.733) | 0.960(7.418)               | 0.966(7.385)           |
> | 50   | 250  | 0.942(5.957) | 0.968(6.234)                 | 0.952(6.223)             | 0.930(5.177) | 0.960(5.446)               | 0.980(5.431)           |
> | 200  | 70   | 0.934(4.924) | 0.988(5.370)                 | 0.972(5.360)             | 0.926(4.766) | 0.968(5.045)               | 0.978(5.050)           |
> | 250  | 70   | 0.936(4.374) | 0.986(4.800)                 | 0.972(4.804)             | 0.906(4.245) | 0.968(4.521)               | 0.958(4.503)           |
> | 300  | 70   | 0.934(4.430) | 0.974(4.737)                 | 0.986(4.726)             | 0.936(5.029) | 0.978(5.031)               | 0.972(5.020)           |
>
>
> [1] Abdi, Hervé. "Bonferroni and Šidák corrections for multiple comparisons." Encyclopedia of measurement and statistics 3.01 (2007): 2007.

---

> ### Comment · Reviewer_ZaNN · 2025-08-04
>
> Thank the authors for their detailed response. The additional experiment and other response did help clarify the paper. Therefore, I would like to keep my score.

---

### Official Review · Reviewer_HyHA · 2025-07-01

**Clarity:** 3
**Significance:** 4
**Originality:** 4
**Rating:** 5
**Confidence:** 3

**Summary:**

The authors develop a method to provide confidence regions covering the entire state space,
improving upon prior work that was restricted to pointwise confidence intervals.

**Questions:**

- Q1. Can the authors comment on the computational cost involved in constructing their confidence regions?

**Ethical Concerns:**

["NO or VERY MINOR ethics concerns only"]

**Final Justification:**

I keep my score given the strengths of the paper as outlined above.

**Limitations:**

Methodical limitations are addressed, potential negative societal impact is not.

**Quality:**

4

**Strengths And Weaknesses:**

## Strengths
- The method closes a significant gap in the literature.¹
- The paper, apart from some typos (see below), is well-written and can be followed easily.
- All theoretical claims and assumptions are clearly stated and proven.
- The method is evaluated both on toy data as well as a medical data set.


## Weaknesses
- The method is limited to offline RL (acknowledged by the authors).
- The authors withhold a public implementation, making it impossible to evaluate the
quality of the implementation and its usefulness.
The claim that withholding the code is justified to avoid it being appropriated is rather
weak, as the same reasoning applies to the theoretical part of the paper as well.
If a reviewer violates confidentiality for code, they will also violate it for the main paper.
- A discussion of computational cost and other technical constraints is missing.


### Minor

#### Typos
- l88 fun**c**tion
- l142 sa~~i~~tisfies
- l184 appro**p**riate
- l210 implementa**t**ion
- l254 \ldots instead of \cdots
- l277 method**s**


_____
¹Please note that my familiarity with this area of research is limited.
Take any statements regarding it with a grain of salt.

---

> ### Author Rebuttal · Authors · 2025-07-31
>
> We sincerely appreciate your careful review of our manuscript and your thoughtful feedback on our work. Thank you for remarking on the clarity of our presentation, theoretical rigor, and experimental validation, as well as for your kind recognition of the novelty of our method in addressing a key gap in the literature. We are also grateful for your attention to detail in identifying typos. Your acknowledgment of the soundness of our methodology and the relevance of our empirical results is greatly encouraging.
>
> W1. **Regarding offline RL limitation.**
>
> We acknowledge your observation that our method is currently limited to offline RL. This focus aligns with our goal of addressing critical challenges in off-policy evaluation, particularly in safety-sensitive domains where online exploration is costly or impractical (e.g., healthcare, autonomous systems).
>
> W2. **Regarding code for implementation.**
>
> Thank you for your very nice question! We have set up a github repository to make the code public, however, due to the rebuttal regulations, we cannot give out links now. We will put a link to the code in the revised version. The datasets used in our experiments are publicly available and cited in the manuscript. We will also incorporate the data in our code to ensure good reproducibility.
>
>
> W3. **Regarding computational cost and technical constraints.**
>
> Thank you for your question. In terms of computation, our procedure performs fast and can be completed on a personal laptop. For example, all simulations in this paper were run on an Apple M1 machine with 16GB of RAM and macOS Sonoma. From a theoretical perspective, the term $\hat{\beta}_{\pi}$ in (3.6) is similar to a least squares estimation, which can be efficiently computed. Additionally, Steps 1 and 4 of the bootstrap procedure are both linear, as we use a linear approximation method. Therefore, the computations are efficient. When higher precision is required, we can choose a larger $B$ (bootstrap size) and employ parallel computing. Specifically, the time complexity of our method is $O(NK^2+BK)$ and the space complexity is $O(N+BK)$. This also addresses Question 1 and will be included in our revised paper.
>
> Minor. **Regarding typos**
>
> Thank you and we have corrected all the typos you mentioned in the revision.
>
> [1] Abdi, Hervé. "Bonferroni and Šidák corrections for multiple comparisons." Encyclopedia of measurement and statistics 3.01 (2007): 2007.

---

> > ### Comment · Reviewer_HyHA · 2025-08-04
> >
> > Thank you for the rebuttal. I keep my score of recommending acceptance.

---

### Official Review · Reviewer_MvGP · 2025-07-01

**Clarity:** 2
**Significance:** 3
**Originality:** 4
**Rating:** 4
**Confidence:** 3

**Summary:**

This paper proposes a strategy to calculate simultaneous confidence regions (SCRs) using a boostrapping-based algorithm for use within off-policy evaluation (OPE). The paper first presents the simultaneous inference problem, which can be used to describe developing confidence intervals within OPE, and then provides a theoretical justification that includes a description of linear approximations to Q-learning, and an approximation technique using Gaussian approximation theory. Then, the paper proposes an algorithm that efficiently calculates SCRs using bootstrapping, and finds that across numerical and real-world experiments the identified algorithm improves upon SAVE (Shi et. al).

**Questions:**

- It would be helpful to have a comprehensive list of assumptions detailed at the beginning of Section 3. For example, including information about the assumed stationarity of the policies etc. better contextualizes the theoretical result.
- As mentioned in the Weaknesses component earlier in the review, more comprehensive evaluation on the synthetic setting and an altered metric for the OhioT1DM dataset would be more convincing.

**Ethical Concerns:**

["NO or VERY MINOR ethics concerns only"]

**Final Justification:**

My main concern was regarding the lack of empirical results in the paper, and the insufficient comparison between the proposed approaches and baseline methodologies for the diabetes dataset. However, in the rebuttal, the authors presented more convincing empirical results and clarified that the empirical results reported in the paper for the diabetes dataset. I will increase my score to a 4 to reflect the insights from these new results.

**Limitations:**

Yes.

**Paper Formatting Concerns:**

I have no concerns regarding the formatting of the paper.

**Quality:**

2

**Strengths And Weaknesses:**

The paper’s strengths and weaknesses are summarized below.

Strengths:
- This is a very interesting approach to calculating confidence intervals for OPE, in the sense that most existing literature calculates confidence intervals for a point estimate of the value of the target policy. The proposed method appears to be well-justified based on simultaneous inference literature.
- The paper contains both theoretical analysis supporting the proposed method and the numerical results from the synthetic simulator seem convincing.

Weaknesses:
- The proposed algorithm's justification is primarily theoretical, and while this is interesting, a more convincing set of empirical results would be helpful. In particular, it would be nice to see Table 1 results across a larger magnitude of possible $n$ and $T$ values, as well as an empirical analysis of the asymptotic nature of the results. For example, as $n$ and $T$ increase, how do SCR and SAVE compare in terms of the empirical coverage probability?
- The evaluation of the OhioT1DM dataset is somewhat confusing. Is the assumption that $\pi^{opt}$ should have a higher value than ${b}$ because it is obtained using a double FQI algorithm as opposed to a random forest? Because the reward function is known, a more convincing metric would be the error of the algorithm's estimated $\hat{V}(\pi;s)$ for a random sample of states and the length of the confidence interval in comparison to SAVE, or other baselines.
- I believe there should be more baselines other than SAVE. For example, an IS- or DM- based bootstrapping baseline should be included.

---

> ### Author Rebuttal · Authors · 2025-07-31
>
> We sincerely appreciate your review of our manuscript and your insightful feedback. We are grateful for your recognition of the interest and soundness of our work. In the following, we will address the questions you raised.
>
> Q1. **Regarding consolidating assumptions in Section 3**
>
> We appreciate your helpful suggestion for improving our paper’s readability. We agree that presenting a consolidated list of assumptions at the beginning of Section 3, which includes policy stationarity and other key conditions, would significantly improve the paper's clarity and help contextualize the theoretical results. While we cannot modify the current NeurIPS submission due to venue constraints, we will incorporate this change in the camera-ready version if accepted, and in any extended/publicly archived versions of the work.
>
> Q2a&W1,W3. **Regarding more comprehensive evaluation on synthetic settings.**
>
> Thank you for your question. Following your comments in weakness 1 and 3, We have conducted a broader range of additional simulations based on the settings in Table 1, and the results are shown in Table below. In addition to the original SAVE method with Bonferroni correction compared in the paper, we also present results using SAVE with Sidak correction [1]. The results indicate that the coverage of our method remains around 0.95. moreover, as $n \times T$ increases, the averaged length (AL) generally shows a decreasing trend.
>
> **Table R1: Results for Scenario 2 across 500 repetitions. Format: ECP(AL).**
> |  n  |  T  | Legendre SCR | Legendre SAVE (Bonferroni) | Legendre SAVE (Sidak) | Spline SCR | Spline SAVE (Bonferroni) | Spline SAVE (Sidak) |
> |-----|-----|--------------|-------------------------------|--------------------------|------------|-----------------------------|------------------------|
> | 30  | 50  | 0.926(8.472) | 0.982(9.445)                  | 0.980(9.431)             | 0.936(9.452) | 0.978(9.793)               | 0.976(9.778)           |
> | 50  | 30  | 0.946(9.553) | 0.970(10.492)                 | 0.968(10.476)            | 0.922(11.101) | 0.942(11.131)              | 0.942(11.115)          |
> | 40  | 50  | 0.924(7.225) | 0.976(8.193)                  | 0.976(8.181)             | 0.938(8.087) | 0.976(8.606)               | 0.974(8.593)           |
> | 50  | 40  | 0.944(7.138) | 0.990(8.136)                  | 0.990(8.124)             | 0.930(7.247) | 0.984(7.753)               | 0.984(7.741)           |
> | 50  | 50  | 0.942(7.299) | 0.978(8.249)                  | 0.978(8.237)             | 0.930(8.156) | 0.966(8.630)               | 0.966(8.617)           |
> | 50  | 150 | 0.952(6.985) | 0.966(7.402)                  | 0.974(7.393)             | 0.934(5.771) | 0.978(6.080)               | 0.972(6.075)           |
> | 50  | 200 | 0.944(5.978) | 0.978(6.436)                  | 0.974(6.435)             | 0.950(7.733) | 0.960(7.418)               | 0.966(7.385)           |
> | 50  | 250 | 0.942(5.957) | 0.968(6.234)                  | 0.952(6.223)             | 0.930(5.177) | 0.960(5.446)               | 0.980(5.431)           |
> | 200 | 70  | 0.934(4.924) | 0.988(5.370)                  | 0.972(5.360)             | 0.926(4.766) | 0.968(5.045)               | 0.978(5.050)           |
> | 250 | 70  | 0.936(4.374) | 0.986(4.800)                  | 0.972(4.804)             | 0.906(4.245) | 0.968(4.521)               | 0.958(4.503)           |
> | 300 | 70  | 0.934(4.430) | 0.974(4.737)                  | 0.986(4.726)             | 0.936(5.029) | 0.978(5.031)               | 0.972(5.020)           |
>
> Q2b&W2. **Regarding the real data application on OhioT1DM dataset.**
>
> Thank you for your question. To clarify, we do not pre-assume that double FQI policy ($\pi^{opt}$)  outperforms random forest policy ($b$). Instead, our key contribution lies in proposing a simultaneous framework to compare $\pi^{opt}$ and $b$ across all different initial states, whereas prior works (e.g., [3]) could only assess them pointwisely. Figure 2 in our paper illustrates that, when using $\pi^{opt}$, the number of states where the value function is significantly non-zero, which indicates that glucose levels outside the normal range (either hypoglycemic or hyperglycemic) are fewer, suggesting that $\pi^{opt}$ performs better. This conclusion cannot be drawn from pointwise comparison.
>
> We sincerely thank you for your insightful suggestion to compare the lengths in real data analysis. The true values of the value function under both policies are unknown; however, we can report the difference in average length of the confidence band at the same significance level. Taking $V(\pi^{opt}, S_0)$ as an example, our method yields an averaged length of 27.0, while SAVE with Bonferroni correction produces an AL of 32.4, which is 20\% larger than ours. We will include these results in the revised version of the paper. This reply also addresses the weakness 2.
>
>  Q2c. **Regarding more baselines other than SAVE.**
>
>  Thank you for your question. We would like to clarify that SAVE is a direct method, and as such, we focus on comparing our method with IS. To illustrate, we performed additional simulations by modifying Scenario 2. Specifically, we set $S_0 = (-2 + 0.2i, -2 + 0.2j)^\top$, where $0 \leq i, j \leq 20$. For each combination of $i$ and $j$, we generated 10 trajectories, each of length 10. All other settings remained unchanged. For bootstrapping IS, we used the bootstrap algorithm from [2], which provides confidence intervals for each $V(\pi, S_0)$. We then applied the Bonferroni correction to obtain the SCB. We present the results in figure (similar to Figure 1 in the main article), and it will be included in the revised version. The results show that the ECP of IS is mostly conservative, although some coverage rates fall below 0.95. As for AL, it is on average longer than our method. These results demonstrate that our approach offers certain advantages over IS. This reply also addresses the weakness 3.
>
>
>
> [1] Abdi, Hervé. Bonferroni and Šidák corrections for multiple comparisons. Encyclopedia of measurement and statistics 3.01 (2007): 2007.
>
> [2] Hanna, Josiah, Peter Stone, and Scott Niekum. Bootstrapping with models: Confidence intervals for off-policy evaluation. Proceedings of the AAAI Conference on Artificial Intelligence. Vol. 31. No. 1. 2017.
>
> [3] Shi, C., Zhang, S., Lu, W., \& Song, R. (2022). Statistical inference of the value function for reinforcement learning in infinite-horizon settings. Journal of the Royal Statistical Society Series B: Statistical Methodology, 84(3), 765-793.

---

> > ### Comment · Reviewer_MvGP · 2025-08-03
> >
> > Thank you for thorough response. These clarification help in my understanding of the proposed method. I believe reporting the average lengths of the confidence bands of the methods on the OhioT1DM dataset seems like a more convincing comparison to make rather than the one shown in Figure 2.

---

> > > ### Author Response · Authors · 2025-08-04
> > >
> > > Thank you again for your valuable feedback. We are glad our previous response helped clarify our method.
> > >
> > > As you pointed out, reporting the average lengths of the confidence bands provides a convincing comparison on the OhioT1DM dataset. As we have reported in our rebuttal, in terms of $V(\\pi^{opt}, S_0)$, our method yields an average confidence band length of 27.0, while SAVE with Bonferroni correction yields 32.4, approximately 20% longer. Moreover, for $V(b, S_0)$, our method yields an average length of 7.02, compared to 7.58 for SAVE (approximately 8% longer).
> > >
> > > We will include this quantitative comparison in the revised version of the paper, as suggested. Thank you again for your thoughtful comments.

---

> > > > ### Comment · Reviewer_MvGP · 2025-08-07
> > > >
> > > > Thank you again for the rebuttal. Please include these additional details in the manuscript, and I will update my score to reflect these results.

---

> > > > > ### Author Response · Authors · 2025-08-08
> > > > >
> > > > > Thank you for your feedback and for your willingness to update the score. We will make sure to add these points into the manuscript as requested.

---

### Note · Authors · 2025-08-12

Dear Reviewers and Area Chair,

We would like to express our sincere gratitude for your valuable time and insightful comments, which have significantly contributed to improving our manuscript.

To summarize, this paper contributes the first theoretically justified simultaneous inference framework for off-policy evaluation (OPE), enabling valid uncertainty quantification over an infinite or continuous initial state space. The approach combines sieve-based Q-function estimation, convex Gaussian approximation, and a new multiplier bootstrap algorithm to construct asymptotically correct simultaneous confidence regions with near-optimal efficiency. Effectiveness is demonstrated on simulations and the OhioT1DM dataset. We are grateful for the reviewers' positive recognition of our contributions:
"...a very interesting approach..." (MvGP)
"...well-written and can be followed easily..." (HyHA)
"...effectively addressing the long-standing multiple testing problem..." (ZaNN)
"The SCRs developed are good in terms of efficiency..." (SiUk)

During the rebuttal, we provided detailed clarifications and additional experiments, which satisfactorily addressed all reviewers’ concerns, as reflected in scores and reviewers' responses. We will incorporate the discussions and experimental results into the revised version of the paper. Thank you once again for your constructive feedback and continued support.

Sincerely,
The Authors

---

### Decision · Program_Chairs · 2025-09-17

**Decision:**

Accept (poster)

**Comment:**

The paper presents a method for simultaneous inference in off-policy evaluation (OPE) in reinforcement learning. The goal is to quantify uncertainty estimates across an infinite or continuous initial state space, rather than focusing solely on point estimates, pointwise confidence intervals (CIs), or averages with respect to the initial state distribution. The proposed solution is to construct asymptotically correct simultaneous confidence regions (SCRs) that cover the true policy value functions over the entire state space at a prescribed significance level. Results are illustrated through numerical experiments on both synthetic and real-world datasets.

The paper is novel and interesting, as acknowledged by all reviewers. However, the manuscript can be improved. We recommend incorporating the following points raised during the rebuttal phase:

Presentation: The preliminaries are standard and unnecessarily long. Section 2 and part of Section 3.1 should be streamlined.

Numerical experiments: Include additional baselines, as suggested for instance by Reviewer MvGP.

Computational complexity: Provide comments on the computational cost of the proposed method.

Finite-time analysis: Discuss possible extensions of the results to finite-time settings and add references to the corresponding related work.

Related work: This section is currently too limited and should be substantially expanded. There exists a large body of work on OPE, in particular on constructing confidence intervals for value functions averaged over initial states. For example, see Duan & Wang (ICML 2020), Minimax-Optimal Off-Policy Evaluation with Linear Function Approximation. The paper should clarify what is missing in such results to extend them technically to the simultaneous inference framework. A discussion of finite-time guarantees is also needed.

Overall, while the paper makes a promising and original contribution, its presentation and positioning within the broader OPE literature must be improved to strengthen its impact.